# TRAM: Bridging Trust Regions and Sharpness Aware Minimization

**Tom Sherborne**[1][*]   **Naomi Saphra**[2]    **Pradeep Dasigi**[3]    **Hao Peng**[4][*]
[1]University of Edinburgh    [2]Kempner Institute, Harvard University    [3]Allen Institute for AI
[4]University of Illinois Urbana-Champaign
`tom.sherborne@ed.ac.uk, nsaphra@fas.harvard.edu`
`pradeepd@allenai.org, haopeng@illinois.edu`

## Abstract

Sharpness-aware minimization (SAM) reports improving domain generalization by reducing the loss surface curvature in the parameter space. However, generalization during *fine-tuning* is often more dependent on the transferability of *representations* in the function space. Trust-region methods (TR) target this goal by regularizing representation curvature to reduce catastrophic forgetting of pre-trained task-agnostic information while adopting task-specific skills. We consider unifying these strategies for low curvature in both parameter space and function space to improve out-of-domain (OOD) generalization. We propose **Trust Region Aware Minimization** (TRAM), a SAM algorithm fine-tuning for low parameter sharpness and smooth, informative representations preserving pre-trained structure. TRAM uses a trust region bound to inform the SAM adversarial neighborhood, introducing an awareness of function curvature within optimization for flatter minima. We empirically validate TRAM in vision (cross-dataset adaptation) and text (OOD language modeling, zero-shot cross-lingual transfer) tasks where robust domain transfer and representation generality are critical. TRAM outperforms SAM- and TR-based optimization across all tasks, notably surpassing competing methods for hard transfer between *anticorrelated* domains. TRAM establishes a novel standard in fine-tuning for domain-generalizable models with minimal additional computation over previous sharpness-aware methods.

## 1 Introduction

Neural model training requires navigating over a complex, non-convex loss surface (Frankle, 2020) towards a good local minimum. Studying loss surfaces and training dynamics has led to many algorithmic advances (Izmailov et al., 2018; Foret et al., 2021; Chen et al., 2023) and regularization schemes (Srivastava et al., 2014; Ioffe & Szegedy, 2015) to improve optimization. One such strategy is to exploit an association between generalization and flat minima, defined by Hochreiter & Schmidhuber (1994) as "region[s] in weight space with the property that each weight vector from that region has [a] similar small error". Intuitively, a flatter, or less sharp (Keskar et al., 2017), minimum will generalize better, as the loss function will be non-increasing under distribution shift. Recent work has developed a family of *sharpness-aware minimization* (SAM) algorithms targeting flat minima by jointly minimizing a worst-case generalization bound and local parameter sharpness (Foret et al., 2021; Kwon et al., 2021; Kim et al., 2022; Möllenhoff & Khan, 2023).

While flat minima methods report widespread improvement over conventional optimizers (Kaddour et al., 2022), we argue that they are not fully connected to the modern *fine-tuning* paradigm, wherein a task-specific model inherits parameters from a pre-trained model instead of being trained from scratch (Wang et al., 2019; Liang et al., 2020). In these settings, focusing on local properties of the loss landscape (e.g., sharpness) may fail by suboptimally exploiting useful generic task-agnostic structures within pre-trained representations. In this work, we propose to combine sharpness-aware minimization with the robust transfer of pre-trained information (in representation space) for fine-tuning scenarios requiring out-of-distribution knowledge for successful adaptation.

---

[*] This work was done while Tom Sherborne and Hao Peng were at the Allen Institute for AI.

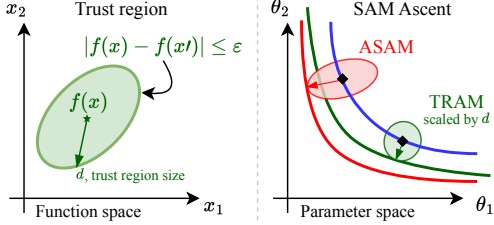

Figure 1: TRAM introduces an awareness of function curvature (i.e., the trust region) into sharpness-aware minimization. (left) TRAM estimates the size of the trust region, $d$, around $f(x)$ in green. (right) the loss contour in parameter space following Kwon et al. (2021) where blue is the typical loss; red is the maximized worst-case loss for ASAM; and green is the maximized loss within the subdomain constrained for function smoothness.

Existing methods to improve leveraging pre-trained structure during fine-tuning include trust region regularization (Schulman et al., 2015; Jiang et al., 2020; Aghajanyan et al., 2021) or adversarial perturbation (Zhu et al., 2020; He et al., 2021). These methods focus on the curvature of the function itself e.g., by encouraging smooth local changes in representations. The intuition is that lower representation curvature during fine-tuning limits a function from catastrophically forgetting (French, 1999, *inter alia*) useful information from pre-training. This representation smoothing approach contrasts with SAM-style optimization for parameter smoothness. Both perspectives show empirical improvement in downstream tasks (Aghajanyan et al., 2021; Bahri et al., 2022), but a fusion of these strategies is presently under-explored.

To this end, we propose **TRAM**: Trust Region Aware Minimization, a fine-tuning algorithm for out-of-distribution generalization combining the success of both sharpness-aware and trust region optimization. TRAM uses a trust region bound to inform the SAM adversarial neighborhood, introducing an awareness of function curvature within optimization for flatter minima. The resulting algorithm yields low-sharpness parameters and improved adaptation of pre-trained models to downstream tasks. To illustrate TRAM's advantage over strong baselines in retaining generic representations, we focus on *distribution transfer* challenges within Transformer-based models. Our contributions are:

- We propose a new optimization algorithm: **Trust Region Aware Minimization** integrates representation smoothing regularization into sharpness-aware minimization. We propose and contrast multiple variants of TRAM based on differing perspectives on trust region estimation and efficiency trade-offs (Section 3).[1]

- We highlight that TRAM empirically improves generalization for multiple out-of-distribution adaptation tasks across vision and natural language: cross-dataset adaptation for image classification, cross-domain language modeling and zero-shot cross-lingual transfer (Section 4).

- We analyze how TRAM limits catastrophic forgetting and optimizes flatter minima to improve fine-tuning. By characterizing major and minor distribution shifts, we identify how TRAM outperforms the trend in anticorrelated generalization scenarios. Our analysis verifies that TRAM optimizes a smoother loss surface for both in-domain and out-of-domain distributions. TRAM also improves representation similarity between seen and unseen distributions to improve cross-domain classification (Section 4).

## 2 BACKGROUND

We describe SAM and trust region optimization, highlighting how these approaches have similar goals. Our motivation for TRAM is the unifying features of each approach outlined in Table 1.

**Notation:** We consider function $f : X \to Y$ parameterized by weights $\theta$ and evaluated by loss function $\ell : Y \times Y \to \mathbb{R}_+$. The expected loss on true distribution $\mathcal{D}$ is $L_{\mathcal{D}}(\theta) = \mathbb{E}_{(x,y) \sim \mathcal{D}}[\ell(y, f(x; \theta))]$ and the empirical estimate is $L_S = \frac{1}{n} \sum_S \ell(y_i, f(x_i; \theta))$ sampling $n$ training samples, $S = \{(x_i, y_i)\}_{i=1}^n$, from $\mathcal{D}$. Functional distance on model outputs is measured by the Kullback-Leibler divergence $D_{\mathrm{KL}}(p||q)$ between target $p$ and estimate $q$. We describe successful domain transfer to distribution $\mathcal{D}'$ as a non-increasing loss for sample $S' \sim \mathcal{D}'$.

**Sharpness-Aware Minimization:** Foret et al. (2021) define local sharpness as $\max_{\|\epsilon\|_2 \leq \rho} L_S(\theta + \epsilon) - L_S(\theta)$. The SAM objective (Equation 1) regularizes parameter magnitude to minimize this sharpness metric jointly with loss within local parameter neighborhood $\rho$.

---

[1]Code at `github.com/tomsherborne/tram_optimizer`.

$$L_S^{\text{SAM}} = \min_\theta \max_{\|\epsilon\|_2 \leq \rho} L_S\left(\theta + \epsilon\right) + \frac{\lambda}{2}\|\theta\|_2^2 \quad (1) \qquad \epsilon_{\text{ASAM}}^* = \rho\,\frac{\theta^2 \nabla L_S}{\|\theta \nabla L_S\|_2} \qquad (2)$$

This min-max optimization problem is solved in alternating stages. Initial ascent perturbs parameters $\theta$ to $\theta + \epsilon$, where $\epsilon$ is a perturbation maximizing loss (to minimize local sharpness). The feasible region for perturbation $\epsilon$ is a Euclidean spherical neighborhood with radius $\rho > 0$. Successive descent evaluates gradients at $\theta + \epsilon$ for gradient descent at $\theta$ using the local worst-case loss.

The optimal $\epsilon$, the perturbation for worst-case loss within the $\rho$-ball, is the source of ongoing debate. Foret et al. (2021) express a closed-form solution setting $\epsilon$ as the radius $\rho$ scaled by the normalized gradient. Kwon et al. (2021) propose Adaptive SAM (ASAM) to improve SAM with invariance to the loss scaling. For ASAM, each parameter within $\theta$ is perturbed by $\rho$ scaled by parameter gradient and the parameter norm (Equation 2). TRAM follows SAM in setting $\epsilon$ with scale invariance and also augments $\epsilon$ such that the update in $\theta$ respects a maximum divergence in the function space.

**Trust Region Regularization:** Trust region regularization encourages low curvature during optimization by regularizing the function output distribution with respect to a previous step's distribution. A fine-tuned model with high curvature (i.e., distance) to pre-trained representations may struggle to connect task-specific knowledge with novel domains. This approach proves successful in penalizing large policy updates in reinforcement learning (Schulman et al., 2015), encouraging local smoothness to adversarial perturbation (Jiang et al., 2020) and minimizing catastrophic forgetting for domain transfer (Aghajanyan et al., 2021).

Equation 3 defines the objective under Trust Region Policy Optimization (TRPO; Schulman et al., 2015) constraining loss, $L_S$, with a regularization term $d_\theta$. TRPO idealizes smoothness in $f\left(x\right)$ by regularizing local function similarity to the previous iterate. The update at $t$ is constrained such that changes in probability density, $p_f\left(\cdot \mid x, \theta\right)$ are no larger than some $\varepsilon$ measured by divergence $d : \mathcal{Y} \times \mathcal{Y} \to \mathbb{R}_+$. There are several ways of defining $d$—we consider options in Equations 4 to 5.

$$\min_\theta \; L_S\left(\theta\right) \text{ subject to } d_\theta \leq \varepsilon \qquad (3)$$

Equation 4 estimates the trust region as the KL divergence between predictive distributions at the previous and current step. Intuitively, penalizing divergence from prior steps encourages the function to stay "close" to the previous distribution i.e., within the trust region of equivalent output. Across training, $d_\theta$ encourages small updates with low curvature between fine-tuned and pre-trained models.

$$d_\theta\left(\theta_{t-1}, \theta_t\right) = \mathbb{E}_{x \sim D}\left[D_{\text{KL}}\left(p_f\left(\cdot \mid x, \theta_{t-1}\right) \| p_f\left(\cdot \mid x, \theta_t\right)\right)\right] \qquad (4)$$

Equation 5 provides the penalty from R3F (Aghajanyan et al., 2021) where $d_x$ estimates the trust region by sampling from inputs under parametric noise. This penalizes the divergence between $p_f\left(\cdot \mid x, \theta_t\right)$ and $p_f\left(\cdot \mid x + z, \theta_t\right)$ for some zero-mean noise $z \sim \mathcal{N}\left(0, \sigma^2\right)$. R3F proposes that sampling $z$ estimates the trust region by simulating a distribution shift in $p_f$ corresponding to perturbed $x + z$. This encourages similarity to a neighborhood around $f\left(x, \theta\right)$ with equivalent output. Either approach estimates the permissible distance for an update in $\theta$ without increasing local representation curvature. We focus on trust region methods to improve generalization across distributions via improved leveraging of pre-trained structure (Jiang et al., 2020, *inter alia*).

$$d_x\left(x + z, x\right) = \mathbb{E}_{z \sim \mathcal{N}}\left[D_{\text{KL}}\left(p_f\left(\cdot \mid x + z, \theta\right) \| p_f\left(\cdot \mid x, \theta\right)\right)\right] \qquad (5)$$

**Comparison:** SAM, TRPO, and R3F have similar goals in searching for generalizable solutions while appearing superficially distinct. We compare the broad motivations and qualities of methods in Table 1, highlighting both perspectives optimize for smoothness in different spaces.

SAM minimizes sharpness within a neighborhood in $\theta$ set by scalar parameter $\rho$. Trust region regularization penalizes loss by scalar distance $d_\theta$ or $d_x$. We hypothesize that this regularization can inform the size of the SAM neighborhood. Can we jointly minimize sharpness and penalize high curvature in representations? Considering the sharpness objective $\left(\max_{\|\epsilon\|_2 \leq \rho} L_S\left(\theta + \epsilon\right) - L_S\left(\theta\right)\right)$ we consider if this $\epsilon$ can also satisfy the Equation 3 constraint of $d_\theta$ or $d_x < \varepsilon$. Our intuition here is to minimize parameter sharpness (i.e., SAM) only within an update promoting low representation curvature. Combining the features of these solutions could improve generalization to unseen distributions during fine-tuning.

Table 1: Comparison between SAM-style, trust region and TRAM learning. SAM optimizes parameters for low sharpness, trust region methods optimize for low-curvature representations. TRAM combines these strategies to bound SAM-style learning within a trust region neighborhood.

| | Goal | $\epsilon$ | Distance | Domain | Gradient | Forward/Backward |
|---|---|---|---|---|---|---|
| SAM-style | Low-sharpness $\theta$ | Equation 2 | — | $\rho$-ball | $\nabla L_S$ at $\theta + \epsilon$ | $2 \rightarrow, 2 \leftarrow$ |
| Trust region | Low-curvature $f(y\|x,\theta)$ | — | $d_\theta$ or $d_x$ | $D_{\text{KL}}$ over Distance | $\nabla L_S + d_\theta$ or $d_x$ | $2 \rightarrow, 1 \leftarrow$ |
| TRAM | Both | Equation 7 | $d_\theta$ or $d_x$ | $d_\theta$- or $d_x$-ball | $\nabla L_S$ at $\theta + \epsilon$ | $3 \rightarrow, 2 \leftarrow$ |

## 3 TRAM: TRUST REGION AWARE MINIMIZATION

We consider methods improving generalization by encouraging low-sharpness parameters and task transfer by encouraging low curvature in representation space. We introduce **TRAM**: **T**rust **R**egion **A**ware **M**inimization unifying sharpness-aware and trust region optimization. Kim et al. (2022) raise that the $\rho$ hyperparameter defining the ascent neighborhood in SAM is an "ad hoc" scaling with little relationship to the loss landscape or parameter geometry. We propose to instead define the ascent region by a trust region in representation space.

TRAM substitutes $\rho$ in Equation 2 with the trust region metric, $d: \mathcal{Y} \times \mathcal{Y} \rightarrow \mathbb{R}_+^*$, as defined in Section 2. We estimate trust regions using the divergence from a prior model distribution ($d_\theta$, Equation 4) or divergence from the current distribution under parametric noise ($d_x$, Equation 5). TRAM constrains the maximization domain for ascent (i.e., $\theta \rightarrow \theta + \epsilon$) to the parameter corollary for the trust region i.e., $\max_{\|\epsilon\|_2 \leq d}$ substituted within Equation 1. TRAM perturbs $\theta$ with a loss perturbation only within the parameter neighborhood constrained for low representation curvature. This introduces function curvature awareness within TRAM in addition to the sharpness-awareness objective for flatter minima. In contrast, the maximization region, $\rho$ in SAM/ASAM has no sensitivity to function curvature. We build TRAM on ASAM, and not SAM, after observing strictly better performance in our preliminary experiments.

$$\nabla L_{\text{TRAM}}(\theta) = \left. \frac{\partial L_S}{\partial \theta} \right|_{\theta = \theta + \epsilon_{\text{TRAM}}^*} \qquad (6) \qquad\qquad \epsilon_{\text{TRAM}}^* = \frac{d\, \theta^2 \nabla L_S(\theta_t)}{\|\theta \nabla L_S(\theta_t)\|_2} \qquad (7)$$

The gradient descent update in TRAM is Equation 6, where $\epsilon_{\text{TRAM}}^*$ is solved as Equation 7 by direct substitution of $\rho$ in ASAM. Algorithm 1 in Appendix B.6 details the full training algorithm for TRAM based on the SAM-style min-max optimization routine. TRAM does not require tuning a $\rho$ hyperparameter for stable training. TRAM using $d_\theta$ introduces no new hyperparameters, and using $d_x$ requires only tuning $\sigma$ for additive noise $z$. We hypothesize that TRAM jointly minimizes parameter sharpness and representation curvature to minimize catastrophic forgetting of pre-trained structure. Our results in Section 4 empirically validate this hypothesis.

**Connection to ASAM:** The geometric interpretation of TRAM frames the maximization domain defined by $d$ as a subdomain of the $\rho$-radius Euclidean ball defined in ASAM. Whereas ASAM defines a fixed radius by $\rho$ at each step, TRAM instead uses the nonzero $d$ radius constraining the maximization domain to additionally satisfy the trust region constraint outlined in Equations 3 to 5. Foret et al. (2021, Theorem 2) defines a PAC-Bayesian generalization bound for SAM on $L_\mathcal{D}$ assuming $\rho > 0$. Kwon et al. (2021, Theorem 3) identify a similarly valid bound when considering the norm-adaptive scaling on $\epsilon_{\text{ASAM}}^*$ as in Equation 2. We assume $d \leq \rho$ for similar asymptotic behavior for $\epsilon_{\text{TRAM}}^*$ to $\epsilon_{\text{ASAM}}^*$. We infer that TRAM inherits the existing generalization bound of ASAM for any $\rho > 0$ directly substituted for $d$ i.e., TRAM is a subsolution of ASAM. We can constrain $d$ such that $\max_{\theta_{\neg t}} d_\theta(\theta_{\neg t}, \theta_t) \leq \rho$ or $\max_z d_x(x + z, x) \leq \rho$ to enforce this bound $d \in (0, \rho]$. However, we empirically observe this constraint is satisfied for the optimal setting of $\rho$ in ASAM.

**Improving Efficiency with TRAM-Fisher:** Kim et al. (2022) propose an alternative to SAM removing the Euclidean assumption for parameter geometry. Fisher SAM (FSAM) instead exploits the *statistical manifold* induced by the Fisher Information metric of predictive distribution of the function, $p_f(y \mid x, \theta)$ (Amari, 1998) to set $\epsilon$. This measures statistical divergence between $\theta$ and $\theta + \epsilon$ resulting in $\epsilon_{\text{FSAM}}^*$ in Equation 8 defining an ellipsoid around $\theta$ scaled by the Fisher Information matrix, $F(\theta)$. $F(\theta)$ is prohibitively expensive at scale and is approximated with Equation 9, the diagonal of the squared gradient sum for each batch $B$.

Table 2: We propose four variants of TRAM based on different trust region estimations. TRAM-$\theta_{t-1}$ uses divergence against the previous step; TRAM-$\theta_0$ is a simplifying heuristic of this divergence against the pre-trained model only; and TRAM-$x$ uses noised input divergence, $d_x$. TRAM-Fisher extends FSAM by measuring the Fisher Information metric around the trust region.

| Variant | Trust region measurement | $\epsilon$ | Domain | Forward/Backward |
|---|---|---|---|---|
| TRAM-$\theta_{t-1}$ | $d_\theta\left(\theta_{t-1}, \theta_t\right)$ | Equation 7 | $d_\theta$-ball | $3 \rightarrow, 2 \leftarrow$ |
| TRAM-$\theta_0$ | $d_\theta\left(\theta_0, \theta_t\right)$ | Equation 7 | $d_\theta$-ball | $3 \rightarrow, 2 \leftarrow$ |
| TRAM-$x$ | $d_x\left(x + z, x\right), z \sim \mathcal{N}\left(0, \sigma^2\right)$ | Equation 7 | $d_x$-ball | $3 \rightarrow, 2 \leftarrow$ |
| TRAM-Fisher | $\hat{F}\left(x + z; \theta\right), z \sim \mathcal{N}\left(0, \sigma^2\right)$ | Equation 8 | $\hat{F}$-ellipse | $2 \rightarrow, 2 \leftarrow$ |

$$\epsilon^*_{\text{FSAM}} = \frac{F(\theta)^{-1}\nabla L_S}{\sqrt{\nabla L_S F(\theta)^{-1}\nabla L_S}} \quad (8) \qquad \hat{F}\left(\theta\right) = \text{Diag}\left(\frac{1}{|B|}\sum_{i \in B}\left(\log p_f(y_i|x_i, \theta)\right)\right)^2 \quad (9)$$

We propose TRAM-Fisher as an efficient variant of TRAM inspired by Fisher SAM. Where FSAM measures the Fisher Information geometry of $\theta$ under input $x$, we instead sample the geometry of $\theta$ under the trust region estimation from $x + z$. Our proposal is minimal: replace $p\left(y_i|x_i, \theta\right)$ with $p\left(y_i|x_i + z_i, \theta\right)$ to estimate the Fisher Information Matrix of the *trust region* neighborhood as $\mathbb{E}_{z \sim \mathcal{N}}\left[\hat{F}\left(x + z; \theta\right)\right]$. We sample parametric noise $\{z_i\}_{i=0}^{|B|}$ identically to TRAM and now scale learning with the information geometry of the low curvature neighborhood, $f\left(x + z\right)$. TRAM-Fisher uses the same number of forward/backward passes as FSAM and only requires additional processing to sample $z$ and compute $x + z$. TRAM-Fisher matches FSAM in runtime efficiency (with marginal additional operations) and performs competitively across our experiments. The full TRAM-Fisher algorithm is shown in Appendix B.6.

**Summary:** We propose three variants of TRAM, and TRAM-Fisher, summarized in Table 2. TRAM-$\theta_{t-1}$ follows TRPO (Schulman et al., 2015) in using previous step parameters, $\theta_{t-1}$, to measure the trust region. We also propose a simplification of TRAM-$\theta_{t-1}$ estimating the trust region using $d_\theta$ between current $\theta_t$ and pre-trained model $\theta_0$. TRAM-$\theta_0$ improves training efficiency by removing an updating $\theta_{t-1}$ state. TRAM-$x$ follows R3F (Aghajanyan et al., 2021) using noise-based trust region measurement with additional hyperparameter $z$ for sampling parametric noise. Practically, TRAM requires one additional forward pass adding marginal overhead to the extant complexity of SAM-style training. Despite this additional cost, Section 4 identifies empirical benefits to TRAM and targeted improvement to out-of-domain loss surface sharpness and cross-domain representation similarity.

We outline our datasets in Appendix A, and experiment design in Appendix B for both vision and language modalities. We compare to gradient descent methods (SGD, Adam), sharpness aware methods (SAM, ASAM, FSAM), and trust region methods (TRPO, R3F, MESA) further detailed in Appendix B.2. Broadly, we investigate the hypothesis that *out-of-distribution generalization improves by jointly minimizing parameter sharpness and representation curvature in the function.*

## 4 RESULTS

### 4.1 CROSS-DATASET IMAGE CLASSIFICATION

First, we validate the performance of TRAM in a standardized setting for comparison to other SAM-style optimizers. We evaluate adapting ViT-base (Dosovitskiy et al., 2021) from ImageNet pre-training to image classification fine-tuning. We follow the setup of Kim et al. (2022, Section 5.1) evaluating adaptation to CIFAR-100 (Krizhevsky, 2009), Stanford Cars (Krause et al., 2013), and Oxford Flowers (Nilsback & Zisserman, 2008). Appendix B.3 details our experiment design.

Table 3 details the Top-1 accuracy results for this experiment with direct comparison to Kim et al. (2022, Table 3). The best-performing variant of TRAM (TRAM-$\theta_{t-1}$ or TRAM-$x$) is significantly superior to the closest FSAM competitor ($p < 0.01$). Other variants of TRAM, TRAM-$\theta_0$ or TRAM-Fisher, are largely competitive with prior methods. Our observations validate the hypothesis that TRAM improves adaptation across datasets during fine-tuning for image classification. This comparison acts as a sanity check and demonstrates the utility of our method compared to other SAM-

Table 3: Cross-dataset adaptation from ImageNet to CIFAR-100, Stanford Cars and Oxford Flowers. We report Top-1 classification accuracy averaged over five runs, $\pm$ the 95% confidence interval, for direct comparison to Kim et al. (2022).

| | CIFAR-100 ($\uparrow$) | Cars ($\uparrow$) | Flowers ($\uparrow$) |
|---|---|---|---|
| SGD | 87.97±0.12 | 92.85±0.31 | 94.53±0.20 |
| SAM | 87.99±0.09 | 93.29±0.01 | 95.05±0.06 |
| ASAM | 87.97±0.08 | 93.28±0.02 | 95.08±0.10 |
| FSAM | 88.39±0.13 | 93.42±0.01 | 95.26±0.03 |
| TRAM-$\theta_{t-1}$ | 88.47±0.16 | 93.49±0.04 | 97.07±0.10 |
| TRAM-$\theta_0$ | 88.31±0.09 | 93.16±0.07 | 95.53±0.10 |
| TRAM-$x$ | 88.78±0.01 | 93.32±0.11 | 96.34±0.03 |
| TRAM-Fisher | 88.02±0.18 | 93.12±0.13 | 94.90±0.11 |

Table 4: M2D2 perplexity (lower is better) on Wikipedia (upper) & S2ORC (lower) splits. TRAM-$\theta_{t-1}$ significantly improves over prior work ($p < 0.01$ Kolmogorov-Smirnov test). Results are grouped as: (i) optimizers; (ii) trust region methods; and (iii) TRAM variants. The leftmost column is the training domain and we evaluate zero-shot perplexity on ten domains unseen during fine-tuning (full details in Appendix A). ZS Avg. is the macro-average of all zero-shot domains.

| **Wiki** | Soc. | Cult. | Gen. | Health. | Hist. | Human. | Math. | Nat. | Phil. | Rel. | Tech. | ZS Avg. $\downarrow$ |
|---|---|---|---|---|---|---|---|---|---|---|---|---|
| GPT-2 | 27.2 | 27.7 | 27.8 | 24.5 | 29.2 | 28.8 | 28.6 | 29.4 | 27.8 | 27.7 | 28.7 | 28.0 |
| Adam | 24.8 | 26.3 | 26.4 | 23.6 | 27.2 | 27.0 | 27.4 | 27.6 | 26.3 | 25.8 | 27.4 | 26.5 |
| SAM | 24.5 | 25.9 | 26.0 | 23.1 | 26.9 | 26.6 | 26.6 | 27.2 | 25.8 | 25.5 | 27.0 | 26.1 |
| ASAM | 24.8 | 25.4 | 25.6 | 22.5 | 27.1 | 26.4 | 26.3 | 26.7 | 25.5 | 25.5 | 28.1 | 25.9 |
| FSAM | 21.7 | 23.0 | 23.3 | 20.6 | 23.9 | 23.7 | 23.8 | 24.0 | 23.1 | 22.8 | 24.0 | 23.2 |
| TRPO | 21.8 | 23.0 | 23.3 | 20.7 | 24.0 | 23.7 | 23.8 | 24.0 | 23.1 | 22.8 | 24.1 | 23.3 |
| R3F | 21.8 | 23.0 | 23.3 | 20.7 | 24.0 | 23.7 | 23.8 | 24.0 | 23.1 | 22.8 | 24.1 | 23.3 |
| MESA | 23.1 | 24.0 | 24.3 | 21.5 | 25.4 | 24.9 | 24.8 | 25.2 | 24.1 | 24.0 | 25.1 | 24.3 |
| TRAM-$x$ | 21.9 | 23.1 | 23.4 | 20.7 | 24.0 | 23.3 | 23.9 | 23.9 | 23.2 | 22.7 | 23.9 | 23.2 |
| TRAM-$\theta_{t-1}$ | 20.9 | 22.4 | 22.7 | 20.1 | 23.1 | 22.9 | 23.2 | 23.3 | 22.4 | 22.0 | 23.4 | 22.5 |
| TRAM-$\theta_0$ | 21.9 | 23.1 | 23.4 | 20.7 | 23.9 | 23.3 | 23.9 | 23.8 | 23.1 | 22.7 | 23.9 | 23.2 |
| TRAM-Fisher | 22.5 | 23.7 | 24.0 | 21.3 | 24.6 | 24.0 | 24.7 | 24.6 | 23.8 | 23.3 | 24.6 | 23.9 |

| **S2ORC** | Math | Art | Astro | CondM. | CS | Econ. | NLin. | Phil. | Phys. | QBio | Stat | ZS Avg. $\downarrow$ |
|---|---|---|---|---|---|---|---|---|---|---|---|---|
| GPT-2 | 27.6 | 35.8 | 32.4 | 30.9 | 27.9 | 29.5 | 27.6 | 33.7 | 33.5 | 30.9 | 23.4 | 30.6 |
| Adam | 11.4 | 44.2 | 33.9 | 20.1 | 21.2 | 21.0 | 14.7 | 41.9 | 29.5 | 30.8 | 16.9 | 27.4 |
| SAM | 10.5 | 45.3 | 33.2 | 18.7 | 20.3 | 20.0 | 13.7 | 42.4 | 28.3 | 30.2 | 16.1 | 26.8 |
| ASAM | 10.3 | 45.6 | 33.2 | 18.5 | 20.1 | 19.8 | 13.5 | 42.6 | 28.2 | 30.2 | 15.9 | 26.8 |
| FSAM | 10.4 | 45.6 | 33.3 | 18.5 | 20.2 | 19.9 | 13.5 | 42.7 | 28.3 | 30.2 | 15.9 | 26.8 |
| TRPO | 10.4 | 46.0 | 33.4 | 18.6 | 20.3 | 20.0 | 13.6 | 42.9 | 28.4 | 30.4 | 16.0 | 26.9 |
| R3F | 10.4 | 46.0 | 33.4 | 18.6 | 20.2 | 20.0 | 13.6 | 42.9 | 28.4 | 30.4 | 16.0 | 26.9 |
| MESA | 11.9 | 44.1 | 34.1 | 20.8 | 21.7 | 21.6 | 15.3 | 41.7 | 30.0 | 31.0 | 17.4 | 27.8 |
| TRAM-$x$ | 10.4 | 44.9 | 33.0 | 18.6 | 20.1 | 19.9 | 13.6 | 42.0 | 28.1 | 30.0 | 15.9 | 26.6 |
| TRAM-$\theta_{t-1}$ | 9.6 | 46.8 | 32.5 | 17.2 | 19.2 | 18.9 | 12.6 | 43.3 | 27.0 | 29.6 | 15.0 | 26.2 |
| TRAM-$\theta_0$ | 10.4 | 44.8 | 33.0 | 18.6 | 20.1 | 19.9 | 13.6 | 42.0 | 28.2 | 30.0 | 15.9 | 26.6 |
| TRAM-Fisher | 10.5 | 46.1 | 32.4 | 18.7 | 20.3 | 20.0 | 13.6 | 43.0 | 28.2 | 30.3 | 16.0 | 26.9 |

style optimizers. TRAM yields improved fine-tuned image classification models by encouraging smoothness in parameter and function space.

## 4.2 Cross-domain language modeling

We now consider zero-shot cross-domain language modeling using the M2D2 Corpus (Reid et al., 2022) outlined in Appendix A. We hypothesize that TRAM can improve domain transfer in language modeling by retaining domain-agnostic information from pre-training when fine-tuning to a specific domain. We train a GPT-2 Base model (Radford et al., 2019) on the largest domain in each split of M2D2 (Soc. domain 379M tokens for Wikipedia and Math 1.4B tokens for S2ORC) and evaluate perplexity across ten domains unseen during fine-tuning. Appendix B.4 details our complete experiment design.

Our results in Table 4 validate our hypothesis for TRAM in the cross-domain setting to improve out-of-domain language modeling fine-tuning on a single domain. All TRAM variants (excluding

Figure 2: Perplexity on S2ORC training domain (MATH) and zero-shot domains. We report perplexity across: (a) domains correlated with MATH as STEM domains (see Appendix C.1), (b) ART domain, and (c) the Philosophy (PHIL.) domain. Each figure includes linear regression trends: the blue dotted trend is for prior work and green dashed line includes all TRAM variants. Positive slope ($\rho > 0$) represents correlated domains, negative slope ($\rho < 0$) represents anticorrelated domains. We report Pearson $\rho$ correlation for the blue trend noting $p < 0.01$ significance.

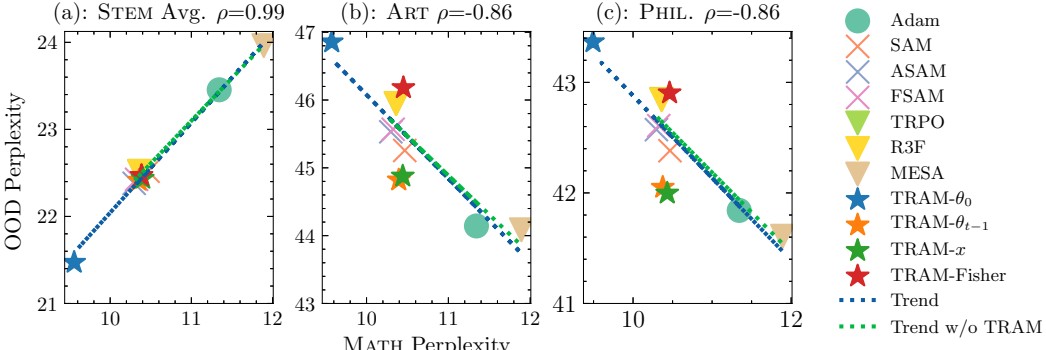

TRAM-Fisher) perform comparably or above competitors in zero-shot transfer across both splits of M2D2. TRAM improves domain transfer in fine-tuned models by better leveraging pre-trained information from unseen domains within the smoother minima idealized by SAM-style training. Generally, the naive Adam baseline or the MESA trust region comparison perform poorest at cross-domain language modeling for Wikipedia or S2ORC splits respectively. As with image classification, FSAM is the strongest competitor to TRAM. The best variant in both splits is TRAM-$\theta_{t-1}$ improving in-domain and average zero-shot perplexity. TRAM-$\theta_{t-1}$ uses the TRPO method of estimating the trust region using the parameters of the previous step. This variant always yields the lowest perplexity in the training domain and the majority of similar and distant zero-shot domains. We additionally verify that TRAM performs competitively at a larger model scale using GPT2-XL (1.5B parameters) in Table 10 in Appendix C.2. We also compare against a naive combination of methods (e.g., ASAM+TRPO) in Appendix C.3.

TRAM improves perplexity for all domains in the Wikipedia split, where all zero-shot domains are positively correlated with the training domain perplexity. However, we observe that perplexity *degrades* for domains distant from the fine-tuning domain in S2ORC (MATH) which benefit less from shared features. Given that neither SAM-style nor trust region methods inverted this anticorrelation trend, it is unsurprising that TRAM follows suit. This confounder results in the overall best model, TRAM-$\theta_{t-1}$, reporting the *worst* performance for the distant domains where the overall poorest model, MESA, reports the *best* performance. We suggest that optimization alone may be insufficient to improve zero-shot domain adaptation for larger distribution shifts. We discuss further the correlation between domain-specific perplexity in Appendix C.1.

### 4.2.1 EASY AND HARD GENERALIZATION

When evaluating performance variation between different distributional shifts—we find that TRAM improves on all prior work for minor shifts (e.g., MATH to Physics/PHYS.) and generally matches or improves on a negative trend for major shifts (e.g., MATH to ART). Discussion of out-of-domain generalization often overlooks differences between major and minor shifts. In practice, in-domain performance has a very different relationship to performance when generalizing to a major domain shift rather than a minor shift. Considering minor distribution shifts, accuracy is strongly correlated on in-domain and out-of-domain datasets (Miller et al., 2021). However, major distribution shifts may lead to scenarios where performance is instead *anticorrelated* with in-domain accuracy (Teney et al., 2022). Considering these scenarios in the S2ORC task, we observe that models trained using TRAM often perform better on new domains than their in-domain performance would predict. Furthermore, TRAM improves perplexity across both minor and major distribution shifts.

Figure 2a shows the close positive correlation between performance on the training domain (MATH) and the average across all other STEM disciplines, considering all optimization approaches. As detailed in Appendix C.1, performance correlates with $\rho > 0.8$ between MATH and each individual

Table 5: XNLI accuracy (higher is better) for training language (EN) and 14 zero-shot target languages summarised by ZS AVG. (key in Appendix A). All TRAM variants significantly outperform other methods ($p < 0.01$ Wilcoxon test). Results are grouped as: (i) optimizers; (ii) trust region methods; and (ii) TRAM variants. We report the mean across 20 seeds with standard deviation in Table 13.

| | EN | AR | BG | DE | EL | ES | FR | HI | RU | SW | TH | TR | UR | VI | ZH | ZS AVG. ↑ |
|---|---|---|---|---|---|---|---|---|---|---|---|---|---|---|---|---|
| Adam | 83.9 | 71.2 | 77.1 | 75.7 | 75.2 | 78.3 | 77.6 | 69.6 | 74.9 | 64.6 | 71.2 | 72.2 | 65.8 | 74.1 | 73.1 | 72.9 |
| SAM | 84.8 | 72.1 | 78.1 | 76.7 | 75.7 | 79.0 | 77.9 | 69.8 | 75.7 | 65.2 | 71.8 | 73.1 | 66.8 | 75.1 | 74.2 | 73.7 |
| ASAM | 85.0 | 72.0 | 78.4 | 76.9 | 76.1 | 79.5 | 78.5 | 70.4 | 76.1 | 65.2 | 72.5 | 73.4 | 66.9 | 75.5 | 74.2 | 74.0 |
| FSAM | 84.7 | 72.2 | 78.1 | 76.9 | 76.0 | 79.3 | 78.4 | 70.0 | 76.1 | 65.1 | 72.2 | 73.0 | 66.8 | 75.3 | 74.2 | 73.8 |
| TRPO | 84.9 | 71.3 | 77.7 | 76.2 | 75.3 | 78.6 | 77.3 | 69.2 | 75.2 | 64.4 | 71.6 | 72.4 | 65.3 | 73.8 | 73.3 | 73.0 |
| R3F | 85.5 | 72.7 | 78.9 | 77.5 | 76.8 | 79.9 | 79.2 | 70.7 | 76.8 | 66.2 | 72.9 | 73.9 | 66.6 | 75.8 | 74.6 | 74.5 |
| MESA | 84.9 | 71.9 | 77.9 | 76.7 | 75.7 | 78.8 | 77.8 | 69.6 | 75.8 | 64.1 | 72.1 | 72.4 | 65.7 | 74.4 | 73.9 | 73.3 |
| TRAM-$x$ | 86.2 | 73.5 | 79.8 | 78.3 | 77.5 | 80.9 | 79.6 | 71.4 | 77.5 | 66.0 | 73.8 | 74.3 | 67.6 | 76.7 | 75.9 | 75.2 |
| TRAM-$\theta_{t-1}$ | 86.2 | 73.1 | 79.5 | 78.2 | 77.0 | 80.2 | 79.7 | 71.5 | 77.5 | 66.4 | 73.3 | 74.2 | 66.5 | 76.7 | 75.8 | 75.0 |
| TRAM-$\theta_0$ | 85.6 | 72.9 | 79.3 | 77.8 | 77.4 | 80.2 | 79.6 | 71.2 | 77.1 | 65.9 | 73.3 | 74.2 | 67.5 | 76.7 | 75.8 | 74.9 |
| TRAM-Fisher | 84.3 | 73.1 | 78.7 | 77.1 | 76.2 | 79.5 | 78.4 | 71.4 | 76.6 | 65.7 | 73.2 | 73.6 | 67.5 | 75.5 | 75.5 | 74.4 |

STEM category. Considering the blue dotted trend for previous optimization methods (excluding TRAM), we see that all TRAM optimizers fall on or marginally below the line. This result suggests that TRAM not only supports in-domain performance but specifically improves generalization to similar domains.

By contrast, we find there is generally a trade-off between performance on MATH and the hardest anticorrelated domains: ART (Figure 2b) and Philosophy (PHIL, Figure 2c). Both TRAM-$x$ and TRAM-$\theta_0$ fall far below the trend for previous algorithms where in-domain improvement worsens out-of-domain perplexity. TRAM not only matches or outperforms existing methods on easier generalization cases, but exhibits a lesser trade-off between easy and hard generalization compared to all previous approaches.

## 4.3 ZERO-SHOT CROSS-LINGUAL TRANSFER

Finally, we now consider if TRAM improves cross-lingual adaptation during monolingual fine-tuning. We adapt a multilingual pre-trained model to an English entailment classification task (NLI) and then evaluate the zero-shot cross-lingual capability for the model to classify entailment from inputs in 14 unseen languages. We hypothesize that TRAM benefits cross-lingual transfer via improved application of multilingual pre-trained information to a task with only English training data. In general, languages closer to English (e.g., French, German) are "easier" for transfer than distant or low-resource languages (e.g., Urdu, Swahili) (Ahmad et al., 2019). An ideal system will produce equivalent cross-lingual transfer for all zero-shot languages. Our complete experiment design is outlined in Appendix B.5. We train an XLM-Roberta-based model (Conneau et al., 2020a) on English MultiNLI (Williams et al., 2018) and report accuracy results for the XNLI cross-lingual entailment benchmark (discussed in Appendix A).

Table 5 highlights that TRAM improves over all competing methods for the cross-lingual transfer objective, similar to our findings for cross-dataset image classification and cross-domain language modeling. Similar to the above tasks, TRAM-$x$ and TRAM-$\theta_{t-1}$ are the best-performing algorithms reporting both the strongest in-domain and average out-of-domain accuracy. Either TRAM variant is the best method across all individual languages. We identify that all methods worsen for languages distant from English in a similar trend to language modeling for anticorrelated domains. However, here TRAM is strictly superior to any other method for both near and distant languages to English. Notably, TRAM-Fisher significantly improves upon FSAM ($p < 0.01$) despite the close similarity in methods. Given the additional forward pass required for TRAM-$x$, TRAM-Fisher represents a better performance-complexity trade-off which is competitive in some tasks. We analyze the loss surface and representation transfer in Table 6 to verify that TRAM extends a low-curvature loss surface and representation smoothness to all zero-shot languages. In Appendix C.5, we train a model using TRAM with alternative distances for trust region measurement to analyze the criticality of using KL divergence. We observe that TRAM is robust to multiple distances with marginal degradation. These results empirically verify our hypothesis that training with complementary SAM-style and trust region methods improves the language transferability of a fine-tuned model.

Table 6: Analysis of (a) $\epsilon$-sharpness and (b) CKA representation similarity for TRAM. We measure each metric using the XNLI validation set and report for the training language (EN) and the zero-shot languages (ZS). We report mean of 20 runs $\pm$ standard deviation across languages and the Pearson correlation between EN and ZS AVG. $\epsilon$-sharpness across runs.

| (a) $\epsilon$-sharpness $\downarrow$ | EN | ZS AVG. | Pearson $\rho$ | (b) CKA $\uparrow$ | EN | ZS AVG. |
|---|---|---|---|---|---|---|
| Adam | 2.16 | 1.98$\pm$ 0.79 | 0.29$\pm$0.20 | Adam | 0.69 | 0.44$\pm$ 0.10 |
| SAM | 1.43 | 3.32$\pm$ 0.96 | 0.26$\pm$0.34 | SAM | 0.69 | 0.42$\pm$ 0.10 |
| ASAM | 2.57 | 2.22$\pm$ 0.79 | 0.38$\pm$0.12 | ASAM | 0.69 | 0.42$\pm$ 0.10 |
| FSAM | 2.34 | 2.62$\pm$ 0.29 | 0.27$\pm$0.71 | FSAM | 0.73 | 0.48$\pm$ 0.10 |
| TRPO | 6.17 | 2.36$\pm$ 1.02 | 0.52$\pm$0.25 | TRPO | 0.70 | 0.45$\pm$ 0.10 |
| R3F | 6.22 | 2.56$\pm$ 1.21 | 0.50$\pm$0.12 | R3F | 0.66 | 0.40$\pm$ 0.10 |
| MESA | 2.76 | 5.48$\pm$ 0.75 | 0.21$\pm$0.25 | MESA | 0.67 | 0.42$\pm$ 0.10 |
| TRAM-$\theta_{t-1}$ | 0.50 | 1.19$\pm$0.38 | 0.60$\pm$0.15 | TRAM-$\theta_{t-1}$ | 0.77 | 0.57$\pm$ 0.10 |
| TRAM-$\theta_0$ | 0.75 | 1.92$\pm$0.24 | 0.58$\pm$0.27 | TRAM-$\theta_0$ | 0.69 | 0.45$\pm$ 0.11 |
| TRAM-$x$ | 0.61 | 1.49$\pm$ 0.49 | 0.75$\pm$0.18 | TRAM-$x$ | 0.75 | 0.54$\pm$ 0.11 |
| TRAM-Fisher | 1.67 | 2.02$\pm$ 0.40 | 0.42$\pm$0.37 | TRAM-Fisher | 0.72 | 0.49$\pm$ 0.10 |

**Loss surface dynamics:** Investigating the loss surface, we test the hypothesis that TRAM leads to flatter minima on both *in-domain* and *out-of-domain* data. We evaluate validation set $\epsilon$-sharpness (Keskar et al., 2017), defined in Appendix B.7, across 20 trained models. We report in-domain (for English) and out-of-domain (zero-shot languages) $\epsilon$-sharpness in Table 6 across TRAM and baselines (omitting models which under-performed). Most methods unsurprisingly demonstrate a lower in-domain sharpness but poorer out-of-domain sharpness. TRAM yields a smoother solution for both the in-domain and out-of-domain regions of the loss surface. We also observe an improved average Pearson correlation (and lower variance) between in-distribution and out-of-distribution sharpness using TRAM. This infers that the relationship between loss surfaces of different distributions is more desirably predictable with TRAM. Notably, other SAM-style methods are *worse than Adam* for out-of-domain sharpness—suggesting that current SAM algorithms (excluding TRAM) are possibly "sharpness-aware" only within the training distribution.

**Representation transfer:** We analyze the similarity of pre-trained and fine-tuned representations for the same setup of XNLI. We hypothesize that if TRAM optimizes within the trust region, pre- and post-fine-tuned representations will be more similar to allow better usage of pre-trained structure. We measure this relationship using CKA similarity (Kornblith et al., 2019) defined in Appendix B.8. Similar to the previous analysis, we observe that TRAM produces representations that are more similar to pre-trained XLM-Roberta representations than any competitor. This applies to both the EN case and the ZS AVG. case, with all other models performing similarly to the Adam baseline. Counterintuitively, trust region methods perform no better than SAM-style methods which do not explicitly target representational similarity. This observation could be related to recent insight into the smoothness side effects of training with SAM (Wen et al., 2023). We additionally raise that neither metric in Table 6 shows a similar trend to our empirical findings—comparisons here do not strictly reflect similar performance variation on specific tasks. Despite empirical improvement, recent work questions if sharpness meaningfully correlates with generalization (Juneja et al., 2023; Andriushchenko et al., 2023). Extending TRAM should further evaluate this relationship and investigate how trust region measurement could inform better predictors of generalization capability.

## 5 CONCLUSION

We present TRAM: **T**rust **R**egion **A**ware **M**inimization. TRAM optimizes for smoothness in both parameter and function spaces to improve domain generalization during fine-tuning. TRAM inherits the capability of SAM to optimize towards flatter minima and integrates trust region awareness to ensure low local curvature between output representations. We evaluate TRAM on out-of-distribution scenarios, where the model must generalize to new distributions unseen during training. In this setup, TRAM proves more effective than SAM-style optimization or trust region methods. Our analysis identifies how TRAM bucks the anticorrelated trend for major distribution shifts, learns a flatter out-of-domain loss surface, and improves representation similarity for data unseen during fine-tuning.

## 6 ACKNOWLEDGMENTS

TS gratefully acknowledges the support of the UK Engineering and Physical Sciences Research Council (grant EP/W002876/1). This work has been made possible in part by a gift from the Chan Zuckerberg Initiative Foundation to establish the Kempner Institute for the Study of Natural and Artificial Intelligence.

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

Table 7: Data splits for M2D2 (Reid et al., 2022) across Wikipedia and S2ORC (Lo et al., 2020). For simplicity, we do not consider the fine-grained subdomains in each domain. All data sourced from Huggingface (`huggingface.co/datasets/machelreid/m2d2`)

| Split | Domain | Abbrev. | Size (Tokens) | Training Domain | Train Tokens | Validation Tokens | Test Tokens |
|-------|--------|---------|---------------|-----------------|--------------|-------------------|-------------|
| Wiki | Culture and the arts | CULT. | 289M | | — | — | 34.33M |
| | General reference | GEN. | 60M | | — | — | 2.38M |
| | Health and fitness | HEALTH. | 116M | | — | — | 6.83M |
| | History and events | HIST. | 226M | | — | — | 11.65M |
| | Human activities | HUMAN. | 343M | | — | — | 12.41M |
| | Mathematics and logic | MATH. | 52M | | — | — | 1.65M |
| | Natural and physical sciences | NAT. | 189M | | — | — | 13.45M |
| | Philosophy and thinking | PHIL. | 165M | | — | — | 2.32M |
| | Religion and belief systems | REL | 64M | | — | — | 5.44M |
| | Society and social sciences | SOC. | 397M | ✓ | 380M | 11.8M | 11.74M |
| | Technology and applied sciences | TECH. | 297M | | — | — | 11.78M |
| S2ORC | Art | ART | 98M | | — | — | 1.06M |
| | Astrophysics | ASTRO | 728M | | — | — | 1.14M |
| | Condensed matter | CONDM. | 688M | | — | — | 1.17M |
| | Computer science | CS | 1.1B | | — | — | 1.17M |
| | Economics | ECON. | 11M | | — | — | 1.16M |
| | Mathematics | MATH | 1.4B | ✓ | 1.1B | 1.46M | 1.40M |
| | Nonlinear sciences | NLIN. | 134M | | — | — | 1.28M |
| | Philosophy | PHIL. | 156M | | — | — | 1.06M |
| | Physics | PHYS. | 737M | | — | — | 1.12M |
| | Quantitative biology | QBIO | 336M | | — | — | 1.08M |
| | Statistics | STAT | 450M | | — | — | 1.19M |

## A  DATA SPLITS

### VISION DATASETS

For vision modality experiments, we evaluate cross-dataset transfer from ImageNet (Deng et al., 2009) to CIFAR-100 (Krizhevsky, 2009), Stanford Cars (Krause et al., 2013), and Oxford Flowers (Nilsback & Zisserman, 2008). We source all datasets from HuggingFace[2] using the default training/testing partitions.

### LANGUAGE DATASETS

We evaluate the M2D2 dataset (Reid et al., 2022) for cross-domain language modeling. M2D2 contains two groups: 11 domains from the S2ORC corpus of ArXiv listings (Lo et al., 2020) and an archive of Wikipedia articles. We train a language model on each split's largest domain and evaluate zero-shot generalization to ten domains unseen during fine-tuning. Evaluation uses token-level perplexity across each domain. Table 7 details the partition sizes (in tokens) for each domain in M2D2.

Zero-shot cross-lingual transfer is evaluated using MultiNLI and XNLI for entailment classification. In this task, a model predicts an entailment label (neutral, entailment, contradiction) between sentence pairs. We use only English language MultiNLI (Williams et al., 2018) for training data and evaluate the trained model on the 14 unseen natural languages in XNLI (Conneau et al., 2018) during test time. These datasets are balanced in label classes and we report accuracy per language in our results. A complete breakdown of partition sizes is shown in Table 8.

## B  ADDITIONAL EXPERIMENTAL DETAILS

### B.1  MODEL TRAINING

We fine-tune each pre-trained model without any freezing or additional task-specific parameters where possible. We also do not explore fine-tuning with low-rank approximations or adapters i.e., 'full fine-tuning'. This setup isolates the contribution of the optimization algorithm over additional capacity in the model. For image classification and cross-lingual entailment classification, we follow

---

[2]`huggingface.co/datasets/cifar100`
`huggingface.co/datasets/Multimodal-Fatima/StanfordCars_train`
`huggingface.co/datasets/nelorth/oxford-flowers`

Table 8: Data splits for XNLI (Conneau et al., 2018). The Training data in English is sourced from the MultiNLI dataset (Williams et al., 2018) with translations provided for XNLI. Model selection during training uses only the English validation data. Validation data for other languages is used to measure $\epsilon$-sharpness in our analysis. We omit data splits not used in this work. All data sourced from HuggingFace (`huggingface.co/datasets/xnli`).

| XNLI | Abbrev. | Train Sentences | Validation Sentences | Test Sentences |
|---|---|---|---|---|
| English | EN | 393K | 2.5K | 5K |
| Arabic | AR | — | 2.5K | 5K |
| Bulgarian | BG | — | 2.5K | 5K |
| German | DE | — | 2.5K | 5K |
| Greek | EL | — | 2.5K | 5K |
| Spanish | ES | — | 2.5K | 5K |
| French | FR | — | 2.5K | 5K |
| Hindi | HI | — | 2.5K | 5K |
| Russian | RU | — | 2.5K | 5K |
| Swahili | SW | — | 2.5K | 5K |
| Thai | TH | — | 2.5K | 5K |
| Turkish | TR | — | 2.5K | 5K |
| Urdu | UR | — | 2.5K | 5K |
| Vietnamese | VI | — | 2.5K | 5K |
| Chinese (Simplified) | ZH | — | — | 5K |

fine-tuning norms and only introduce a new dataset-specific 'head' to predict dataset-specific logits. For language tasks, we fine-tune each pre-trained model for 50,000 steps using an initial learning rate of $2 \times 10^{-5}$, a polynomial decay schedule, and 10,000 step learning rate warmup. We use Adam (Kingma & Ba, 2017), with a decay factor setting $(\beta_1, \beta_2) = (0.9, 0.99)$, as the base optimizer for each SAM-style and TR method unless mentioned otherwise. When using validation loss for model selection, we use only the validation partition of the training domain to reflect a stricter evaluation setup without access to additional domains during training. All models are trained $1 \times$A100 80GB GPU for under 72 hours except for GPT2-XL experiments in Appendix C.2.

## B.2 BASELINES

We compare to a naive SGD baseline for vision experiments following Kim et al. (2022). Our naive baseline for language experiments is Adam (Kingma & Ba, 2017) without any augmentation setting decay factors as $(\beta_1, \beta_2) = (0.9, 0.99)$. All algorithms listed below use Adam as the inner optimizer for the final update (e.g., Algorithm 1 Step 6).

For sharpness-aware methods: we compare to SAM ($\rho = 0.05$, Foret et al., 2021), Adaptive SAM (ASAM, $\rho = 0.5$, Kwon et al., 2021) and Fisher SAM (FSAM, $\gamma = 0.1$, $\eta = 0.1$, Kim et al., 2022).

For trust region methods: we compare to Trust Region Policy Optimization (TRPO, Schulman et al., 2015), R3F ($\sigma = 0.1$, Aghajanyan et al., 2021), and MESA (Du et al., 2022). MESA is a variant of TRPO regularizing output representation divergence between current $\theta_t$ and the exponential moving average of previous $\theta_{<t}$ with decay factor 0.999. For trust-region methods, we add the regularizer directly to the task-specific loss function with a weighting coefficient of $\lambda = 0.1$ (in Equation 3).

## B.3 CROSS-DATASET TRANSFER FOR IMAGE CLASSIFICATION

We implement the same cross-dataset adaptation setup as Kim et al. (2022) as a 'sanity check' directly comparing TRAM to prior methods in the same setting. This setup is not strictly similar to the 'out-of-distribution' scenario we report for language tasks—this experiment verifies that TRAM is performant on standard benchmarks and valuably evaluates TRAM in the vision modality. The objective is to adapt ViT-base (Dosovitskiy et al., 2021) from ImageNet pre-training (Deng et al., 2009) to additional image classification tasks. We evaluate dataset adaptation to CIFAR-100 (Krizhevsky, 2009), Oxford Flowers (Nilsback & Zisserman, 2008) and Stanford Cars (Krause et al., 2013) datasets. Our hypothesis is that TRAM can improve applying information from ImageNet to additional datasets with different labels and input data.

---

**Algorithm 1** Trust Region Aware Minimization

---

**Input:** Training set $S = \{(x_i, y_i)\}$, loss function $\ell$, learning rate $\alpha$, model parameters $\theta$, noise standard deviation $\sigma$ {if noise-estimated trust region}.

**for** $t = 1, 2, \ldots$ **do**

    (1) Sample batch of $B = \{(x_i,\ y_i)\}_{i=0}^{|B|}$ data from $S$.

    (2) Compute the predictive distribution, $p_f\left(\cdot | x_B, \theta_t\right)$, and gradient of the batch loss $\nabla L_B(\theta)$.

    (3) Compute trust region distance $d$ as:

        $d_\theta$ using $p_f\left(\cdot | x_B, \theta_{t-1}\right)$ (Equation 4) or

        $d_x$ using $p_f\left(\cdot | x_B + z, \theta_t\right),\ z \sim N\left(0,\ \sigma^2\right)$ (Equation 5).

    (4) Compute $\epsilon^*_{TRAM}$:

        $\epsilon^*_{\text{TRAM}} = d\,\theta^2 \nabla L_S(\theta_t) / \|\theta \nabla L_S(\theta_t)\|_2$

    (5) Ascent step perturbing $\theta$ to $\theta + \epsilon^*_{TRAM}$.

    (6) Compute gradient at $\theta + \epsilon^*_{TRAM}$ as Equation 6:

$$\nabla L_{\text{TRAM}}\left(\theta\right) = \left.\frac{\partial L_S}{\partial \theta}\right|_{\theta = \theta + \epsilon^*_{\text{TRAM}}}$$

    (7) Gradient descent update: $\theta \leftarrow \theta - \alpha \nabla L_{\text{TRAM}}(\theta)$.

**end for**

---

We match the experimental setting of Kim et al. (2022): fine-tuning ViT-base-16 for 200 epochs with a base optimizer of SGD, an initial learning rate of $5 \times 10^{-4}$, and a cosine learning rate decay with no warmup or restarts. We do not use early stopping to match prior work and use the final model regardless of validation loss. We report the average Top-1 accuracy over 5 runs, $\pm$ the 95% confidence interval, in Table 3 for direct comparison to Kim et al. (2022, Table 3).

## B.4 CROSS-DOMAIN LANGUAGE MODELING

We consider zero-shot cross-domain language modeling using the M2D2 Corpus (Reid et al., 2022). Our hypothesis is that TRAM can better apply language modeling information from large text corpora to improve out-of-domain perplexity when fine-tuning to a specific domain. For S2ORC, we train on the "Math" domain (MATH, 1.4B tokens) and for Wikipedia, we train on the "Society and social sciences" domain (SOC., 379M tokens). We use the 112M parameter GPT-2 base model (Radford et al., 2019) with a batch size of 16 blocks of 1024 tokens following the setup of prior work (Reid et al., 2022; Chronopoulou et al., 2022; 2023). We evaluate generalization via perplexity for each test domain. We also evaluate a zero-shot baseline (i.e., GPT-2 before fine-tuning) to contrast with the same model before domain-specific adaptation. To reduce computation, we train one model with one random seed per algorithm.

## B.5 ZERO-SHOT CROSS-LINGUAL TRANSFER

We test zero-shot cross-lingual transfer by fine-tuning a multilingual model on an English task and then evaluating the model in other languages. We hypothesize that TRAM can improve task transfer across languages by improving the usage of information from multilingual pre-training during monolingual fine-tuning. A poorer model may 'forget' other languages during the adaptation process. We evaluate transfer from English to additional languages by predicting labels for the XNLI test set after training the model for NLI only in English. We use the 250M XLM-Roberta Base multilingual pre-trained model (Conneau et al., 2020a) with a classification head trained from scratch. This model uses a batch size of 32 examples using only English validation loss for model selection. Each reported result is averaged across 20 runs of varying random seeds to control for variation in loss surface.

## B.6 TRAINING ALGORITHMS

The training algorithm for TRAM is outlined in Algorithm 1 using different metrics for trust region estimation, $d$, outlined in Section 3. Algorithm 2 details the TRAM-Fisher algorithm. Practically, this modifies Algorithm 1 in removing one forward pass to estimate the trust region distance and instead approximate the Fisher Information Matrix of the trust region neighborhood in representation space.

---

**Algorithm 2** Trust Region Aware Minimization with Fisher Information Matrix (TRAM-Fisher)

---

**Input:** Training set $S = \{(x_i, y_i)\}$, loss function $\ell$, learning rate $\alpha$, model parameters $\theta$, noise standard deviation $\sigma$

**for** $t = 1, 2, \ldots$ **do**

   1) Sample batch of $B = \{(x_i, \ y_i)\}_{i=0}^{|B|}$ data from $S$.

   2) Compute the predictive distribution, $p_f(\cdot|x_B, \theta_t)$, and gradient of the batch loss $\nabla L_B(\theta)$.

   3) Sample input noise $z \sim N\left(0, \ \sigma^2 I_{|\theta|}\right)$.

   4) Approximate the Fisher Information Matrix at $x + z$:

$$\hat{F}(x + z; \ \theta) = \text{Diag}\left(\frac{1}{|B|} \sum_{i \in B} \left(\log p_f(y_i|x_i + z_i, \theta)\right)\right)^2$$

   5) Compute $\epsilon^*_{\text{TRAM-F}}$:

$$\epsilon^*_{\text{TRAM-F}} = \frac{\hat{F}(x+z; \ \theta)^{-1} \nabla L_S}{\sqrt{\nabla L_S \hat{F}(x+z; \ \theta)^{-1} \nabla L_S}}.$$

   6) Ascent step perturbing $\theta$ to $\theta + \epsilon^*_{TRAM-F}$.

   7) Compute gradient at $\theta + \epsilon^*_{TRAM-F}$ as Equation 6:

$$\nabla L_{\text{TRAM-F}}(\theta) = \left.\frac{\partial L_S}{\partial \theta}\right|_{\theta = \theta + \epsilon^*_{\text{TRAM}}}$$

   8) Gradient descent update: $\theta \leftarrow \theta - \alpha \nabla L_{\text{TRAM-F}}(\theta)$.

**end for**

---

### B.7 MEASURING SHARPNESS

We follow Keskar et al. (2017) in evaluating model $\epsilon$-sharpness as Equation 10 where $\ell$ is the loss function, $x \in \mathbb{R}^n$ are $n$ model parameters, $A \in \mathbb{R}^{n \times p}$ is a matrix restricting the $\epsilon$-sharpness to a subspace of $p$ parameters ($A^+$ is the pseudo-inverse of $A$) and $\mathcal{C}_\varepsilon$ is defined as Equation 11 denoting a "box" region around the solution over which loss is maximized.

$$\phi_{x,f}(\epsilon, A) := \frac{\max_{y \in \mathcal{C}_\epsilon} \ell(x + Ay) - \ell(x)}{1 + \ell(x)} \times 100 \tag{10}$$

$$\mathcal{C}_\epsilon = \{z \in \mathbb{R}^p : -\epsilon(|(A^+x)_i| + 1) \leq z_i \leq \epsilon(|(A^+x)_i| + 1) \ \forall \ i \in [p]\} \tag{11}$$

For our measurement of $\epsilon$-sharpness, we set $A$ to the identity matrix $I_{n \times n}$ to measure over the complete model. We measure $\epsilon$-sharpness over the validation set of XNLI in all languages comparing between original loss $\ell(x)$ and maximized loss $\max_{y \in \mathcal{C}_\epsilon} \ell(x + Ay)$. We follow the $\epsilon$-sharpness setup of Juneja et al. (2023) using an SGD optimizer, learning rate of $8 \times 10^{-5}$, a 32 example batch size, accumulation over 4 steps and $\epsilon$ of $1 \times 10^{-5}$.

### B.8 MEASURING REPRESENTATION SIMILARITY

We follow Kornblith et al. (2019) and Conneau et al. (2020b) in evaluating cross-lingual similarity using Centered Kernel Alignment (CKA). At a language level, CKA computes a similarity score between matrix $X$ and $Y$ where $X, Y \in \mathbb{R}^{n \times d}$ are dense matrices of $n$ outputs of $d$-dimensional representations from each model. We compute linear CKA similarity as Equation 12 using the Frobenius norm. For our cross-lingual transfer experiments, we use the base model output for each example (i.e., the representation before the classification head) to evaluate similarity.

$$\text{CKA}(X, Y) = \frac{\|Y^T X\|_F^2}{\|X^T X\|_F \|Y^T Y\|_F} \tag{12}$$

## C ADDITIONAL RESULTS

### C.1 DOMAIN CORRELATIONS FOR S2ORC

Table 9 details the correlation between zero-shot and training domain perplexity across methods. We omit the combination approaches (e.g., ASAM+R3F) due to poor performance. For Wikipedia, all domains are correlated with the training domain indicating that the domain-specific fine tuning on

Table 9: Pearson correlation between training domains and zero-shot domains for M2D2. We report how the change in training domain correlates with changes in zero-shot perplexity to analyze how different domains improve or worsen during fine-tuning. All domains are correlated with SOC. for the Wikipedia split. ART and PHIL. domains are anti-correlated with MATH training domain for S2ORC indicating a major distribution shift.

| Wiki Domain | $\rho$ to SOC. | $p < 0.01$? | S2ORC Domain | $\rho$ to MATH | $p < 0.01$? | STEM? |
|---|---|---|---|---|---|---|
| CULT. | 0.982 | ✓ | ART | -0.861 | ✓ | |
| GEN. | 0.983 | ✓ | ASTRO | 0.812 | ✓ | ✓ |
| HEALTH. | 0.970 | ✓ | CONDM. | 0.999 | ✓ | ✓ |
| HIST. | 0.998 | ✓ | CS | 0.996 | ✓ | ✓ |
| HUMAN. | 0.980 | ✓ | ECON. | 0.997 | ✓ | ✓ |
| MATH. | 0.976 | ✓ | NLIN. | 1.000 | ✓ | ✓ |
| NAT. | 0.982 | ✓ | PHIL. | -0.825 | ✓ | |
| PHIL. | 0.985 | ✓ | PHYS. | 0.991 | ✓ | ✓ |
| REL | 0.994 | ✓ | QBIO | 0.932 | ✓ | ✓ |
| TECH. | 0.983 | ✓ | STAT | 0.998 | ✓ | ✓ |
| ZS AVG. | 0.990 | ✓ | ZS AVG. | 0.968 | ✓ | |
| | | | STEM AVG | 0.998 | ✓ | |

Table 10: M2D2 perplexity across training algorithms for GPT2-XL. We fine-tune on the MATH domain M2D2 S2ORC split and evaluate in-domain and out-of-domain perplexity. We evaluate TRAM, competitive comparisons and a GPT2-XL zero-shot baseline. We omit algorithms demonstrating poorer results in smaller scale experiments to limit computation demands. As in Table 4, TRAM performs strongly compared to all comparisons. We report the average zero-shot perplexity (ZS AVG.) as the summary metric to judge domain transfer capability (lower is better). Worst perplexity (excluding zero-shot) is red, best is green.

| S2ORC | MATH | ART | PHIL. | ASTRO | CONDM. | CS | ECON. | NLIN. | PHYS. | QBIO | STAT | ZS AVG. ↓ |
|---|---|---|---|---|---|---|---|---|---|---|---|---|
| GPT2-XL | 16.9 | 22.8 | 21.2 | 19.8 | 19.0 | 17.3 | 18.5 | 17.8 | 20.7 | 19.8 | 14.9 | 19.2 |
| Adam | 8.7 | 30.4 | 28.2 | 24.0 | 14.9 | 15.4 | 15.4 | 11.4 | 21.4 | 22.1 | 12.6 | 19.6 |
| SAM | 8.7 | 29.3 | 28.0 | 22.6 | 14.6 | 15.1 | 15.1 | 11.2 | 20.5 | 21.4 | 12.3 | 19.0 |
| ASAM | 7.9 | 28.0 | 26.1 | 21.8 | 13.4 | 14.1 | 14.1 | 10.4 | 19.4 | 20.3 | 11.4 | 17.9 |
| FSAM | 7.8 | 26.7 | 25.0 | 21.1 | 13.1 | 13.7 | 13.7 | 10.2 | 18.8 | 19.6 | 11.2 | 17.3 |
| TRPO | 8.9 | 27.9 | 26.4 | 23.0 | 14.9 | 15.3 | 15.3 | 11.5 | 20.8 | 21.3 | 12.5 | 18.9 |
| R3F | 8.9 | 27.9 | 26.4 | 23.0 | 14.9 | 15.3 | 15.3 | 11.5 | 20.8 | 21.3 | 12.5 | 18.9 |
| MESA | 9.1 | 28.7 | 26.7 | 23.7 | 14.8 | 15.0 | 16.3 | 13.1 | 20.7 | 23.2 | 12.8 | 19.5 |
| TRAM-$\theta_{t-1}$ | 8.3 | 25.2 | 23.8 | 20.1 | 13.7 | 14.0 | 14.2 | 10.7 | 18.9 | 19.4 | 11.5 | 17.2 |
| TRAM-$x$ | 8.3 | 25.3 | 23.8 | 20.2 | 13.8 | 14.1 | 14.2 | 10.8 | 19.0 | 19.5 | 11.6 | 17.2 |

SOC. domain has a net positive improvement on all zero-shot domains. This trend is not consistent for S2ORC where we observe that ART and PHIL. domains are anti-correlated with the MATH training domain. Improvement to MATH perplexity worsens the performance on these domains across all methods. As discussed in Section 4.2.1, TRAM reports perplexity below this trend to perform better than expected for a negatively correlated trend. For comparison, we contrast the correlations between positively correlated domains (grouped as an average entitled STEM) and anticorrelated domains in Figure 2.

## C.2  TRAINING GPT2-XL WITH TRAM

Bahri et al. (2022) report that training with SAM is effective over all sizes of T5 (Raffel et al., 2020). We verify if this improvement trend extends to TRAM by training a GPT2-XL model (1.5B parameters) on the same language modeling task for 100,000 steps. The setup is the same as described in Appendix B but we use 4 A100 GPUs for training each with a batch size per device of 4 blocks × 1024 tokens. Perplexity for S2ORC domains is shown in Table 10 where we observe similar trends to the 112M parameter GPT2 model. We choose not to run these larger experiments on methods with poor performance in Table 4 (e.g., combined approaches, TRAM-Fisher) to limit computation

demands. Zero-shot GPT2-XL is a stronger baseline here which some methods struggle to improve upon despite improvement in the training domain. TRAM-$\theta_{t-1}$ and TRAM-$x$ perform similarly reporting the lowest perplexity in four domains. The most competitive adjacent algorithm is FSAM reporting the lowest perplexity in seven domains. The difference between FSAM and either TRAM algorithm is not significant here, as we observed for smaller models in Table 4.

## C.3 RESULTS FROM COMBINING OPTIMIZATION ALGORITHMS

Table 11: M2D2 perplexity (lower is better) on Wikipedia (upper) & S2ORC (lower) splits. TRAM-$\theta_{t-1}$ significantly improves over prior work ($p < 0.01$ Kolmogorov-Smirnov test). Results are grouped as: (i) optimizers; (ii) trust region methods; (iii) combined SAM optimizers and trust region methods; and (iv) TRAM variants. The leftmost column is the training domain and we evaluate zero-shot perplexity on ten domains unseen during fine-tuning (full details in Appendix A). ZS Avg. is the macro-average of all zero-shot domains.

| Wiki | Soc. | Cult. | Gen. | Health. | Hist. | Human. | Math. | Nat. | Phil. | Rel. | Tech. | ZS Avg. ↓ |
|---|---|---|---|---|---|---|---|---|---|---|---|---|
| GPT-2 | 27.2 | 27.7 | 27.8 | 24.5 | 29.2 | 28.8 | 28.6 | 29.4 | 27.8 | 27.7 | 28.7 | 28.0 |
| Adam | 24.8 | 26.3 | 26.4 | 23.6 | 27.2 | 27.0 | 27.4 | 27.6 | 26.3 | 25.8 | 27.4 | 26.5 |
| SAM | 24.5 | 25.9 | 26.0 | 23.1 | 26.9 | 26.6 | 26.6 | 27.2 | 25.8 | 25.5 | 27.0 | 26.1 |
| ASAM | 24.8 | 25.4 | 25.6 | 22.5 | 27.1 | 26.4 | 26.3 | 26.7 | 25.5 | 25.5 | 28.1 | 25.9 |
| FSAM | 21.7 | 23.0 | 23.3 | 20.6 | 23.9 | 23.7 | 23.8 | 24.0 | 23.1 | 22.8 | 24.0 | 23.2 |
| TRPO | 21.8 | 23.0 | 23.3 | 20.7 | 24.0 | 23.7 | 23.8 | 24.0 | 23.1 | 22.8 | 24.1 | 23.3 |
| R3F | 21.8 | 23.0 | 23.3 | 20.7 | 24.0 | 23.8 | 23.8 | 24.0 | 23.1 | 22.8 | 24.1 | 23.3 |
| MESA | 23.1 | 24.0 | 24.3 | 21.5 | 25.4 | 24.9 | 24.8 | 25.2 | 24.1 | 24.0 | 25.1 | 24.3 |
| ASAM+TRPO | 25.6 | 26.8 | 26.9 | 24.0 | 28.0 | 27.6 | 27.6 | 28.2 | 26.8 | 26.5 | 27.9 | 27.0 |
| ASAM+R3F | 25.0 | 26.0 | 26.2 | 23.2 | 27.4 | 26.9 | 26.8 | 27.4 | 26.1 | 25.9 | 27.1 | 26.3 |
| ASAM+MESA | 25.3 | 26.3 | 26.5 | 23.5 | 27.7 | 27.2 | 27.1 | 27.7 | 26.3 | 26.1 | 27.4 | 26.6 |
| TRAM-$\theta_{t-1}$ | 20.9 | 22.4 | 22.7 | 20.1 | 23.1 | 22.9 | 23.2 | 23.3 | 22.4 | 22.0 | 23.4 | 22.5 |
| TRAM-$\theta_0$ | 21.9 | 23.1 | 23.4 | 20.7 | 23.9 | 23.3 | 23.9 | 23.8 | 23.1 | 22.7 | 23.9 | 23.2 |
| TRAM-$x$ | 21.9 | 23.1 | 23.4 | 20.7 | 24.0 | 23.3 | 23.9 | 23.9 | 23.2 | 22.7 | 23.9 | 23.2 |
| TRAM-Fisher | 22.5 | 23.7 | 24.0 | 21.3 | 24.6 | 24.0 | 24.7 | 24.6 | 23.8 | 23.3 | 24.6 | 23.9 |

| S2ORC | Math | Art | Astro | CondM. | CS | Econ. | NLin. | Phil. | Phys. | QBio | Stat | ZS Avg. ↓ |
|---|---|---|---|---|---|---|---|---|---|---|---|---|
| GPT-2 | 27.6 | 35.8 | 32.4 | 30.9 | 27.9 | 29.5 | 27.6 | 33.7 | 33.5 | 30.9 | 23.4 | 30.6 |
| Adam | 11.4 | 44.2 | 33.9 | 20.1 | 21.2 | 21.0 | 14.7 | 41.9 | 29.5 | 30.8 | 16.9 | 27.4 |
| SAM | 10.5 | 45.3 | 33.2 | 18.7 | 20.3 | 20.0 | 13.7 | 42.4 | 28.3 | 30.2 | 16.1 | 26.8 |
| ASAM | 10.3 | 45.6 | 33.2 | 18.5 | 20.1 | 19.8 | 13.5 | 42.6 | 28.2 | 30.2 | 15.9 | 26.8 |
| FSAM | 10.4 | 45.6 | 33.3 | 18.5 | 20.2 | 19.9 | 13.5 | 42.7 | 28.3 | 30.2 | 15.9 | 26.8 |
| TRPO | 10.4 | 46.0 | 33.4 | 18.6 | 20.3 | 20.0 | 13.6 | 42.9 | 28.4 | 30.4 | 16.0 | 26.9 |
| R3F | 10.4 | 46.0 | 33.4 | 18.6 | 20.2 | 20.0 | 13.6 | 42.9 | 28.4 | 30.4 | 16.0 | 26.9 |
| MESA | 11.9 | 44.1 | 34.1 | 20.8 | 21.7 | 21.6 | 15.3 | 41.7 | 30.0 | 31.0 | 17.4 | 27.8 |
| ASAM+TRPO | 13.7 | 46.6 | 36.9 | 23.6 | 23.8 | 23.8 | 17.4 | 43.8 | 33.1 | 33.5 | 19.2 | 30.2 |
| ASAM+R3F | 13.5 | 46.2 | 36.5 | 23.3 | 23.6 | 23.5 | 17.2 | 43.4 | 32.7 | 33.2 | 19.0 | 29.9 |
| ASAM+MESA | 13.4 | 45.9 | 36.3 | 23.1 | 23.4 | 23.3 | 17.0 | 43.2 | 32.5 | 33.0 | 18.9 | 29.7 |
| TRAM-$\theta_{t-1}$ | 9.6 | 46.8 | 32.5 | 17.2 | 19.2 | 18.9 | 12.6 | 43.3 | 27.0 | 29.6 | 15.0 | 26.2 |
| TRAM-$\theta_0$ | 10.4 | 44.8 | 33.0 | 18.6 | 20.1 | 19.9 | 13.6 | 42.0 | 28.2 | 30.0 | 15.9 | 26.6 |
| TRAM-$x$ | 10.4 | 44.9 | 33.0 | 18.6 | 20.1 | 19.9 | 13.6 | 42.0 | 28.1 | 30.0 | 15.9 | 26.6 |
| TRAM-Fisher | 10.5 | 46.1 | 32.4 | 18.7 | 20.3 | 20.0 | 13.6 | 43.0 | 28.2 | 30.3 | 16.0 | 26.9 |

Given that TRAM builds on integrating SAM-style optimization with trust-region regularization, we additionally compare to a naive combination of each of these methods. We replace the standard loss function in ASAM with the loss function adding trust region regularization.

Our full results featuring these systems are shown in Table 11 for language modeling and Table 12 for zero-shot cross-lingual transfer. Across both tasks, naive combination approaches are some of the weakest approaches. When we directly combine ASAM with each trust region regularizer (TRPO, R3F, MESA), we find that the naive combination approaches perform worse than Adam alone, even with extensive hyperparameter tuning. We conjecture that the constituent methods fail to compound constructively because the trust region regularizer does not interact with (or respect) the $\rho$-ball neighborhood of ASAM. Therefore, each component may contribute to cross-feature interference, with a disadvantageous net effect on training. TRAM instead offers to combine strategies with complementary features without interference.

Table 12: XNLI accuracy (higher is better) for training language (EN) and 14 zero-shot target languages summarised by ZS AVG. (key in Appendix A). All TRAM variants significantly outperform other methods ($p < 0.01$ Wilcoxon test). Results are grouped as: (i) optimizers; (ii) trust region methods; (iii) combined SAM optimizers and trust region regularization; and (iv) TRAM variants. We report the mean across 20 seeds with standard deviation in Table 13.

| | EN | AR | BG | DE | EL | ES | FR | HI | RU | SW | TH | TR | UR | VI | ZH | ZS AVG ↑ |
|---|---|---|---|---|---|---|---|---|---|---|---|---|---|---|---|---|
| Adam | 83.9 | 71.2 | 77.1 | 75.7 | 75.2 | 78.3 | 77.6 | 69.6 | 74.9 | 64.6 | 71.2 | 72.2 | 65.8 | 74.1 | 73.1 | 72.9 |
| SAM | 84.8 | 72.1 | 78.1 | 76.7 | 75.7 | 79.0 | 77.9 | 69.8 | 75.7 | 65.2 | 71.8 | 73.1 | 66.8 | 75.1 | 74.2 | 73.7 |
| ASAM | 85.0 | 72.0 | 78.4 | 76.9 | 76.1 | 79.5 | 78.5 | 70.4 | 76.1 | 65.2 | 72.5 | 73.4 | 66.9 | 75.5 | 74.2 | 74.0 |
| FSAM | 84.7 | 72.2 | 78.1 | 76.9 | 76.0 | 79.3 | 78.4 | 70.0 | 76.1 | 65.1 | 72.2 | 73.0 | 66.8 | 75.3 | 74.2 | 73.8 |
| TRPO | 84.9 | 71.3 | 77.7 | 76.2 | 75.3 | 78.6 | 77.3 | 69.2 | 75.2 | 64.4 | 71.6 | 72.4 | 65.3 | 73.8 | 73.3 | 73.0 |
| R3F | 85.5 | 72.7 | 78.9 | 77.5 | 76.8 | 79.9 | 79.2 | 70.7 | 76.8 | 66.2 | 72.9 | 73.9 | 66.6 | 75.8 | 74.6 | 74.5 |
| MESA | 84.9 | 71.9 | 77.9 | 76.7 | 75.7 | 78.8 | 77.8 | 69.6 | 75.8 | 64.1 | 72.1 | 72.4 | 65.7 | 74.4 | 73.9 | 73.3 |
| ASAM+TRPO | 85.0 | 72.4 | 78.5 | 77.2 | 76.4 | 79.7 | 78.9 | 70.4 | 76.4 | 65.3 | 72.4 | 73.2 | 66.8 | 75.7 | 74.6 | 74.1 |
| ASAM+R3F | 85.1 | 72.1 | 78.3 | 76.9 | 75.9 | 79.3 | 78.4 | 70.3 | 76.0 | 65.1 | 71.8 | 73.0 | 66.3 | 75.1 | 74.3 | 73.8 |
| ASAM+MESA | 84.7 | 71.7 | 77.8 | 76.3 | 75.7 | 78.8 | 77.9 | 69.5 | 75.4 | 64.1 | 71.6 | 72.7 | 65.6 | 74.3 | 73.4 | 73.2 |
| TRAM-$\theta_{t-1}$ | 86.2 | 73.1 | 79.5 | 78.2 | 77.0 | 80.2 | 79.7 | 71.5 | 77.5 | 66.4 | 73.3 | 74.2 | 67.5 | 76.7 | 75.8 | 75.0 |
| TRAM-$\theta_0$ | 85.6 | 72.9 | 79.3 | 77.8 | 77.4 | 80.2 | 79.6 | 71.2 | 77.1 | 65.9 | 73.3 | 74.2 | 67.5 | 76.7 | 75.8 | 74.9 |
| TRAM-$x$ | 86.2 | 73.5 | 79.8 | 78.3 | 77.5 | 80.9 | 79.6 | 71.4 | 77.5 | 66.0 | 73.8 | 74.3 | 67.6 | 76.7 | 75.9 | 75.2 |
| TRAM-Fisher | 84.3 | 73.1 | 78.7 | 77.1 | 76.2 | 79.5 | 78.4 | 71.4 | 76.6 | 65.7 | 73.2 | 73.6 | 67.5 | 75.5 | 75.5 | 74.4 |

Table 13: Standard deviation of accuracy for the XNLI dataset across 20 training runs with varying random seed. Results are split into groups for: (i) optimizers, (ii) trust region methods, (iii) combined methods, (iv) TRAM variants and (v) TRAM using $d_x$ with varying metrics for computing divergence. This accompanies Table 5 and Table 14 which report average values across seeds.

| | EN | BG | DE | EL | AR | ES | FR | HI | RU | SW | TH | TR | UR | VI | ZH |
|---|---|---|---|---|---|---|---|---|---|---|---|---|---|---|---|
| Adam | 0.34 | 0.42 | 0.65 | 0.47 | 0.50 | 0.39 | 0.51 | 0.51 | 0.51 | 0.65 | 0.40 | 0.41 | 0.43 | 0.51 | 0.55 |
| SAM | 0.24 | 0.31 | 0.35 | 0.34 | 0.31 | 0.32 | 0.49 | 0.33 | 0.50 | 0.36 | 0.40 | 0.35 | 0.44 | 0.39 | 0.39 |
| ASAM | 0.33 | 0.33 | 0.45 | 0.39 | 0.47 | 0.36 | 0.51 | 0.44 | 0.44 | 0.51 | 0.56 | 0.42 | 0.47 | 0.45 | 0.50 |
| FSAM | 0.35 | 0.31 | 0.51 | 0.56 | 0.35 | 0.37 | 0.47 | 0.41 | 0.45 | 0.53 | 0.37 | 0.39 | 0.44 | 0.35 | 0.38 |
| TRPO | 0.24 | 0.35 | 0.37 | 0.34 | 0.30 | 0.39 | 0.34 | 0.46 | 0.40 | 0.38 | 0.36 | 0.34 | 0.53 | 0.38 | 0.34 |
| R3F | 0.34 | 0.40 | 0.44 | 0.38 | 0.35 | 0.35 | 0.43 | 0.42 | 0.46 | 0.41 | 0.41 | 0.35 | 0.43 | 0.39 | 0.47 |
| MESA | 0.34 | 0.34 | 0.44 | 0.24 | 0.40 | 0.52 | 0.37 | 0.67 | 0.43 | 0.26 | 0.40 | 0.45 | 0.59 | 0.43 | 0.34 |
| ASAM+TRPO | 0.30 | 0.28 | 0.36 | 0.29 | 0.26 | 0.28 | 0.35 | 0.34 | 0.44 | 0.36 | 0.39 | 0.38 | 0.34 | 0.32 | 0.32 |
| ASAM+R3F | 0.32 | 0.45 | 0.45 | 0.40 | 0.46 | 0.36 | 0.49 | 0.53 | 0.50 | 0.52 | 0.40 | 0.49 | 0.60 | 0.45 | 0.49 |
| ASAM+MESA | 0.34 | 0.31 | 0.30 | 0.44 | 0.42 | 0.39 | 0.38 | 0.51 | 0.58 | 0.46 | 0.44 | 0.31 | 0.51 | 0.50 | 0.46 |
| TRAM-$\theta_{t-1}$ | 0.40 | 0.31 | 0.40 | 0.30 | 0.36 | 0.31 | 0.43 | 0.50 | 0.53 | 0.48 | 0.43 | 0.34 | 0.36 | 0.49 | 0.42 |
| TRAM-$\theta_0$ | 0.34 | 0.38 | 0.41 | 0.44 | 0.48 | 0.40 | 0.43 | 0.53 | 0.63 | 0.47 | 0.66 | 0.39 | 0.57 | 0.50 | 0.54 |
| TRAM-$x$ | 0.31 | 0.29 | 0.45 | 0.44 | 0.37 | 0.33 | 0.38 | 0.37 | 0.48 | 0.39 | 0.44 | 0.32 | 0.43 | 0.39 | 0.42 |
| TRAM-Fisher | 0.29 | 0.65 | 0.67 | 0.55 | 0.58 | 0.60 | 0.49 | 0.72 | 0.69 | 0.64 | 0.86 | 0.49 | 0.68 | 0.55 | 0.73 |
| TRAM-$x$ (MMD) | 0.42 | 0.38 | 0.46 | 0.43 | 0.47 | 0.35 | 0.42 | 0.43 | 0.48 | 0.44 | 0.59 | 0.49 | 0.36 | 0.37 | 0.59 |
| TRAM-$x$ ($L_2$) | 0.30 | 0.27 | 0.27 | 0.26 | 0.24 | 0.28 | 0.27 | 0.21 | 0.29 | 0.26 | 0.24 | 0.28 | 0.21 | 0.22 | 0.22 |

## C.4 RUN VARIATION IN CROSS-LINGUAL TRANSFER

For XNLI experiments, we report the mean over 20 runs varying random seed in Table 5 and Table 14. We report the respective standard deviation values for each reported mean in Table 13.

## C.5 CHOOSING A DISTANCE METRIC

TRAM relies on KL divergence to estimate the trust region around the pre-trained function (i.e., $p_f(\cdot|x + z, \theta)$ or $p_f(\cdot|x, \theta_{t-1})$). We propose TRAM with forward KL on the intuition that the perturbed distribution (i.e., estimated point in the trust region) is the target (i.e., true) output which the current outputs (i.e., estimate) should match. We empirically verify this setup as the optimal arrangement (i.e., forward KL). While reverse KL or symmetric KL report only marginally poorer results, we report only forward KL for simplicity. We also consider alternative distance metrics in Table 14. We evaluate modifying the best-performing model for XNLI with different distances to examine if the divergence for trust region estimation is influential in performance. We evaluate maximum mean discrepancy using an inverse multiquadratic kernel (MMD; Gretton et al., 2012), or $L_2$ distance within $d_x$. Even using the worst-performing metric, $L_2$ distance, TRAM is still

Table 14: XNLI accuracy across varying the divergence metric estimating the trust region distance in TRAM. We compare to using maximum mean discrepancy (MMD) and $L_2$ distance. TRAM is generally robust to different estimates for the trust region between $p_f(\cdot|x,\theta)$ and $p_f(\cdot|x+z,\theta)$.

| | En | Bg | De | El | Ar | Es | Fr | Hi | Ru | Sw | Th | Tr | Ur | Vi | Zh | ZS Avg. ↑ |
|---|---|---|---|---|---|---|---|---|---|---|---|---|---|---|---|---|
| TRAM-$x$ (KL) | 86.2 | 79.8 | 78.3 | 77.5 | 73.5 | 80.9 | 79.6 | 71.4 | 77.5 | 66.0 | 73.8 | 74.3 | 67.6 | 76.7 | 75.9 | 75.2 |
| TRAM-$x$ (MMD) | 86.0 | 79.3 | 78.1 | 77.1 | 73.2 | 80.7 | 79.6 | 71.4 | 77.3 | 66.0 | 74.0 | 74.4 | 67.2 | 76.3 | 75.6 | 75.0 |
| TRAM-$x$ ($L_2$) | 85.1 | 78.7 | 76.8 | 76.2 | 72.2 | 79.4 | 78.8 | 70.4 | 76.2 | 65.5 | 72.6 | 73.5 | 67.1 | 75.8 | 74.6 | 74.1 |

competitive to methods in Table 8. Characterizing the best trust region estimate for TRAM is outside the scope of this work. Future work should explore the suitability of different distances (e.g., Renyi divergence) to improve the estimation of the trust region space.

