# OpenReview forum: "TRAM: Bridging Trust Regions and Sharpness Aware Minimization"
_ICLR.cc/2024/Conference — ICLR 2024 spotlight_

### Official Review · Reviewer_cqYa · 2023-10-27

**Soundness:** 2 fair
**Presentation:** 2 fair
**Contribution:** 2 fair
**Rating:** 5
**Confidence:** 4

**Summary:**

This paper considered adopted trust regions and sharpness aware minimization. Specifically, the gradient update is based on two key steps. Equation (4) illustrates the update rule by KL divergence fine-tuning. Equation (5) illustrates the update of the input by adding a gaussian random variable. Then the model is further validated in GPT2 on different dataset benchmarks.

****Post-Rebuttal****

I would appreciate the hard work and additional experiments by authors. I increased the soundness score.
However, I still feel very confused about several core arguments and positioning within the paper.

- If this paper is a pure empirical paper, this paper should be revised in a major form to highlight the practical contribution rather than new algorithmic contribution in **optimization**. I do not think the contribution is significant in the context of optimization theory.
- If current paper aims to tackle an optimization problem, I do not think the entire analysis is rigorous from a theoretical perspective. I read the related works by authors and I still could not understand why this kind of optimization could have better transferability. Please take note the mentioned theoretical papers did not discuss this point, but rather on the generalization property. (They are different concepts in theory. One is IID and another is OOD). Based on the unclear fundamental points, I could not increase my score, despite numerous additional experiments.

**Strengths:**

- This paper considered an improved method in TRPO. Through smoothly updating the gradient, this method seems to transfer the information.
- Empirical validation in the foundation model and large scale NLP dataset are done.

**Weaknesses:**

- I could not understand *Why* such as transfer could encourage a better transfer. When it does not work? I would like to see a *rigor* mathematical analysis. The sharpness aware minimization could achieve better generalization. How is this mathematically to ensure a better transfer?
I noticed the experiments illustrated
> domain transfer in fine-tuned models by better leveraging the pre-trained structure from unseen domains within the smoother minima idealized by SAM-style training.

However, without clear analysis. This reviewer feels quite difficult to understand why SAM could achieve this objective. Does this approach only work for the selected dataset? **When it fails?**

- I would like to see a computational/memory complexity analysis. How it compares with other methods.

- This paper proposes a general machine learning method while it is only validated in the NLP dataset. Unless the author clearly revised the title and contributions, I would like to see the results in other modalities such as image.

- Equation (3) is not clearly defined. What does it mean by d_{\theta, x}? It is not a rigorous expression.
- Eq(4), Eq(5) why forward KL divergence is considered? Why not reverse KL divergence? Or Other general forms such as Renyi divergence?
- In eq(5), how important is the noise variable? Does the variance of the noise matter?
- Eq(12) may not be effective to correctly estimate the similarity in high dimensional regime. I think there is a complexity issue here (this is not a sample efficient estimator). I could think this value does not make sufficient sense to me in a high-dimensional case.
- Table 4 Why only accuracy is considered a metric? Is this dataset balanced?

**Questions:**

I noticed the primary domain is about optimization.  While there are so many missing points in terms of rigorous analysis in the optimization. If this paper is an applied NLP paper, the paper should be revised in a major form.

---

> ### Author Response · Authors · 2023-11-22
> **Response Part 1**
>
> We thank the reviewer for their time and attention in reviewing our work. We hope our additional results and insight will consider the reviewer in revising their score.
>
> **Q1. On formal analysis**
>
> Please see our response below adapting ViT-base-16 to CIFAR-100 / Cars / Flowers dataset for validation that TRAM also is applicable for computer vision tasks similar to [1,2,3]. We also address that TRAM requires some initial pre-training for applicability in fine-tuning. We can therefore broadly state that TRAM does not work training from scratch at this stage.
>
> We did not discuss the generalization bound of TRAM as we generally consider TRAM as a subsolution to ASAM [2]. The generalization bound of ASAM is described as Theorem 3 from [2] and also derived formally in [4]. On the assumption that $d$ is normally distributed, we expect the generalization bound of TRAM to converge to the same bound as ASAM. ASAM already describes a bound valid for any $\rho>0$ where we ensure that the TR distance is similarly nonzero.
>
> [1] Sharpness-Aware Minimization for Efficiently Improving Generalization https://arxiv.org/pdf/2010.01412.pdf
> [2] ASAM: Adaptive Sharpness-Aware Minimization for Scale-Invariant Learning of Deep Neural Networks https://arxiv.org/pdf/2102.11600.pdf
> [3] Fisher SAM: Information Geometry and Sharpness Aware Minimisation https://proceedings.mlr.press/v162/kim22f/kim22f.pdf
> [4] The intriguing role of module criticality in the generalization of deep networks https://arxiv.org/abs/1912.00528
>
> **Q2. Complexity**
>
> We generally address this by discussing the number of forward and backward passes required for each algorithm (Table 2). The complexity of SAM / ASAM is not discussed in detail by the original authors, similarly referring to the number of forward and backward passes [1,2,3].
>
> At a high level, SAM-style training also needs to permute each parameter in the network.  Permuting each parameter requires some operations to permute $\theta$ to $\theta + \epsilon$. This scales with the number of parameters $N$ with some $k$ operations per parameter (e.g., to compute Equation (2) / Equation (7)).  Computing $\epsilon^{\ast}$ (e.g., Eq(2) or (7)) requires one computation of the model parameter global norm, $g$, for all algorithms. ASAM / TRAM requires one matrix-matrix product to compute $\theta^2$ per parameter —we denote $M^{3}$ for worst case complexity of this operation. TRAM does not require additional computation here as the distance replaces the scalar $\rho$.
>
> We can approximate this as:
>
> 	SGD: 1 forward pass, 1 backward pass
> 	SAM: 2 forward pass, 2 backward pass. $Nk + g$ operations for permuting $\theta$
> 	ASAM: 2 forward pass, 2 backward pass. $N(k + M^{3}) + g$ operations for permuting $\theta$
> 	TRAM: 3 forward pass, 2 backward pass. $N(k + M^{3}) + g$ operations for permuting $\theta$ + constant operation to compute the TR distance.
>
> We empirically observe that the 2 backward passes dominate the learning process as the complexity bottleneck. TRAM has minimal additional overhead beyond ASAM. The permutation operations for $\theta$ to $\theta + \epsilon$ can be parallelized to reduce wall clock runtimes. The memory overhead of TRAM is minimal compared to SAM or ASAM. There is no gradient passed through the permuted parameters for estimating the TR distance, and the final TR distance is a scalar replacing an existing hyper-parameter. Our empirical conclusion is that TRAM has minimal (<10%) increase in wall clock runtime and no additional memory requirements beyond what is required for ASAM.
>
> [1] https://arxiv.org/abs/2010.01412
> [2] https://arxiv.org/pdf/2102.11600.pdf
> [3] https://arxiv.org/abs/2206.04920

---

> > ### Author Response · Authors · 2023-11-22
> > **Response Part 2**
> >
> > **Q3: Vision Modality**
> >
> > For comparison, we implement the same experiments as [1] for FisherSAM training ViT-base-16 [2] for CIFAR-100 [3], Flowers [4] and Cars [5]. We fine-tune ViT-base-16 for 200 epochs training with TRAM and report the Top-1 accuracy(+ 95% confidence interval) averaged over 5 runs  to compare directly to Table 3 in [1]. We match the hyperparameter setup of [1] — the base optimizer is SGD with an initial learning rate of 5e-4 and a cosine decay LR schedule. Due to computational constraints, we only have the bandwidth to report the two main variants of TRAM on these tasks during the rebuttal. We plan to update with the other variants of TRAM (TRAM-$d_{\theta_0}$ and TRAM-Fisher) in future revisions.
> >
> > |           | SGD        | SAM        | ASAM       | FSAM       | TRAM-$d_{\theta_{t-1}}$ | TRAM-$d_{x}$ |
> > |-----------|------------|------------|------------|------------|-------------------------|--------------|
> > | CIFAR-100 | 87.97±0.12 | 87.99±0.09 | 87.97±0.08 | 88.39±0.13 | 88.47±0.16              | **88.78**±0.01   |
> > | Cars      | 92.85±0.31 | 93.29±0.01 | 93.28±0.02 | 93.42±0.01 | **93.49**±0.04              | 93.32±0.11   |
> > | Flowers   | 94.53±0.20 | 95.05±0.06 | 95.08±0.10 | 95.26±0.03 | **97.07**±0.10              | 96.34±0.03   |
> >
> >
> > We observe that TRAM performs competitively across all datasets, with one or both variants of TRAM performing above all other methods. The largest improvement for TRAM is Flowers where we perform +1.81% above FSAM. The smallest improvement is for Cars where we perform 0.07% above FSAM, but we not that the confidence intervals for these results do not overlap.
> >
> > [1] Fisher SAM: Information Geometry and Sharpness Aware Minimisation https://proceedings.mlr.press/v162/kim22f/kim22f.pdf
> > [2] An Image is Worth 16x16 Words: Transformers for Image Recognition at Scale https://arxiv.org/abs/2010.11929
> > [3] Learning Multiple Layers of Features from Tiny Images https://www.cs.toronto.edu/~kriz/learning-features-2009-TR.pdf
> > [4] C3D Object Representations for Fine-Grained Categorization http://vision.stanford.edu/pdf/3drr13.pdf
> > [5] Automated Flower Classification over a Large Number of Classes https://ieeexplore.ieee.org/document/4756141
> >
> > **Q4: Eq. 3**
> >
> > We intend the subscript of $d$ in Eq (3) as a catch-all description of either Eq (5) or Eq (6). We will revise this in future versions.
> >
> > **Q5: KL Divergence**
> >
> > We propose TRAM with forward KL on the intuition that the perturbed distribution is the target which the current outputs should match. Similar to prior work, we do not pass gradients through the perturbed distribution. We note that TRPO uses the same KL formulation but R3F uses a symmetric KL distribution.
> > We examined different KL directions in our preliminary experiments and observed that forward KL was always superior to either symmetric or reverse KL. We agree that other (better) divergences/distances would be valuable within TRAM, however, we consider this out of scope for this paper.
> >
> > XNLI Zero-Shot average accuracy for different KL divergences for TRAM-$d_{x}$ and  TRAM-$d_{\theta_{t-1}}$. Forward refers to the perturbed weights / prior weights producing the target distribution and the current model produces the estimate distribution.
> >
> > | KL Divergence  | TRAM-$d_{x}$ | TRAM-$d_{\theta_{t-1}}$ |
> > |------------|--------------|-------------------------|
> > | Symmetric: | 74.9         | 74.8                    |
> > | Forward:   | 75.2         | 75.0                    |
> > | Reverse:   | 74.7         | 74.7                    |
> >
> > **Q6: Noise variance**
> >
> > Please see our response to MZmv Q4 discussing hyperparameter settings and noise variance.

---

> > > ### Author Response · Authors · 2023-11-22
> > > **Response Part 3**
> > >
> > > **Q7 On CKA**
> > >
> > > We appreciate the concern about the suitability of CKA [1] to estimate high dimensional similarity. We use CKA for appreciable comparison to other work evaluating cross-lingual representation similarity from other methods [2,3,4]. CKA as a similarity index has been extensively used to analyze representations from multilingual BERT and XLM-R [2] on the basis that CKA is useful for datasets and languages with fewer datapoints than the dimensionality ($d=1024$ here and in prior work). CKA is also invariant to orthogonal transformations and isotropic scaling [3] — this is useful in a cross-lingual context as many language subspaces are approximately isotropic [4] but occupy different global regions. This has motivated prior study to use CKA comparing all samples of each language within a dataset
> > >
> > > We appreciate that this answer may be unsatisfactory as the reviewer’s inquiry focuses on if this is a suitable estimator at the given dimensionality. We provide the metrics in Table 5(b) for a direct relationship to other work evaluating cross-lingual similarity. However, we are more than happy to discuss if there exists a better similarity measurement we can consider in future revisions.
> > >
> > > [1] Kornblith et al. (2019) Similarity of Neural Network Representations Revisited
> > >
> > > [2] Conneau et al. (2020) Emerging Cross-lingual Structure in Pretrained Language Models
> > > https://aclanthology.org/2020.acl-main.536/
> > >
> > > [3] Phang et al.  (2021) Fine-Tuned Transformers Show Clusters of Similar Representations Across Layers https://aclanthology.org/2021.blackboxnlp-1.42.pdf
> > >
> > > [4] Rajee et al. (2022) An Isotropy Analysis in the Multilingual BERT Embedding Space https://aclanthology.org/2022.findings-acl.103.pdf
> > >
> > > **Q8: XNLI Statistics**
> > >
> > > We report accuracy for the XNLI dataset as the most common metric for comparison to other research in cross-lingual generalization. Additionally, XNLI test and validation sets are balanced with 33.3% in each class of 'entailment', 'neutral' and 'contradiction' labels. The training split is approximately balanced with a distribution between labels of 33.3332%, 33.3329%,  33.3339%. We are happy to discuss other indexes to measure and report performance improvement from training with TRAM.
> > >
> > > ---
> > > We hope these experiments provide additional utility to the benefits of TRAM. We ask the reviewer to consider revising their score if these additional responses have sufficiently addressed your questions.

---

### Official Review · Reviewer_2Ruf · 2023-10-30

**Soundness:** 2 fair
**Presentation:** 2 fair
**Contribution:** 3 good
**Rating:** 6
**Confidence:** 4

**Summary:**

In this paper, the authors aim to propose a SAM variant to contribute to the training in the area of model fine-tuning, where they propose the Trust Region Aware Minimization. In the method, specific distance measures in TRR are employed as the neighbourhood radius in SAM rather than the manually-set pre-defined radius. The authors claim that the proposed method can optimizer for informative representations without forgetting pre-trained structure. And the authors investigate the perplex on M2D2 Corpus with GPT2 to show the effectiveness of the proposed method.

**Strengths:**

**Strengths**

1. The paper is clearly written and easy to follow.
2. I think the paper aims to contribute to SAM from a very interesting perspective, i.e. fine-tuning techniques. Considering that fine-tuning has become a nearly necessary procedure in NLP tasks, the paper may provide some promising instructions further.
3. Combine the proposed method with Fisher-SAM can reduce extra forward-propagation count when implementing to the same count as in vanilla SAM.

**Weaknesses:**

**Weakness**

1. The core of this proposed method is to adaptively change the neighbourhood radius in SAM (or ASAM) based on certain distance measure. This somehow does not follow the idea of Trust Region Regularization which adds additional constraint on top of the loss according to the measure. More accurately, they are two different things. And, I could not find a clear meaning why using such a distance as the neighbourhood radius could give the "Trust". Several questions arise: what does the "trust" indicate in the proposed method? Why we should not trust the region that is not in the proposed method but in SAM (and ASAM, Fisher-SAM)? Why the given region would not harm the pre-trained models while SAM could? Clear answers are missing in the current paper. Also, it is highly recommended that the authors use figures to illustrate this and the core of the presented method.

2. The Stochastic Weight Averaging (SWA) could also lead to a similar effect as TTR methods. The authors may need to also consider or compare SWA with the TRR and the proposed method. The following papers may be helpful.

    [1] Kaddour, Jean, et al. "When do flat minima optimizers work?." Advances in Neural Information Processing Systems 35 (2022): 16577-16595.

    [2] Wortsman, Mitchell, et al. "Model soups: averaging weights of multiple fine-tuned models improves accuracy without increasing inference time." International Conference on Machine Learning. PMLR, 2022.

3. From the paper, I see nearly no discussions regarding why and how the proposed method could contribute the fine-tuning in the related section, given that the authors claim "their method could not forget the pre-trained structure". BTW, I think their abstract may be somewhat over-claimed. It is interesting to see that the authors are aiming to study the effect of SAM specifically in fine-tuning. But unfortunately, the current version could not present sufficient helpful insights.

4. It would be more impactful that the the authors present results using the proposed method on some recent popular scalable pre-trained llm such as llama.


5. It is highly recommended that the authors release their code.

**Questions:**

See weakness.

**Details Of Ethics Concerns:**

I have not found any discussions about the limitations and potential negative societal impact. But in my opinion, this may not be a problem, since the work only focuses on the learning method in machine learning. Still, it is highly encouraged to add corresponding discussions.

---

> ### Author Response · Authors · 2023-11-21
> **Response Part 1A**
>
> We thank the reviewer for their time and attention in considering TRAM. We provide the first part of our response to address your questions. We are still working on experiments for the vision domain which we hope will provide further insight into TRAM for the reviewer. We will update our response when these are finalized.
>
> **Q1: the notion of "Trust" in the model.**
>
> We thank the reviewer for highlighting where the explanation of trust regions can be improved in the paper. We will expand on this here and include this in future revisions to the paper. We will also improve the visual/graphical description of the method in future revisions.
>
> The notion of the trust region is the local neighborhood around the current function where we can assume an adequately similar representation of the objective function [1,2]. Prior neural TR methods will include this constraint as a direct regularizer on the loss [2,3] without access to the representational probability density [2]. The brief interpretation of this constraint is to iteratively encourage smooth updates to the objective which retain local function similarity to the previous iterate. This regularizer translates the TR objective constraining large changes in the function space into an adjustment to the change in the parameters for a given learning step (i..e, a constraint in $\delta f$ translated into $\delta \theta$). The prior work inspiring TRAM proposes two efficient strategies to approximate the region around the current objective function [2,3] and translating this to a penalty on $\theta$. In either case (i.e., TRAM using Eq (4) or (5)), we introduce a measurement on change in the function space to inform a change in $\theta$.
>
> Our connection to SAM [4] is based on the similarity in geometry between defining the aforementioned change in $\theta$, permitting safe change in the function space, and the $\rho$-ball domain containing sharpness maximization. As noted in [5], the domain of the SAM $\rho$-ball is somewhat arbitrary in that the setting of $\rho$ has little relationship to the parameter geometry or training dynamics. FSAM [4] reformulates SAM to respect the parameter geometry. However, in TRAM we instead propose to define this maximization domain by the space in $\theta$ which permits only “trusted” change in the function. The connection to “trust” in TRAM is to reinterpret what is originally a regularizer in [3,4] to define a search space where SAM-style optimization is permitted. SAM-type algorithms do not contain this notion of “trust”, as the maximization domain has no relationship to the function space and cannot ensure similar smooth changes in function density (e.g., there is no control for staying close to the original space within SAM/ASAM). Therefore, training with SAM does not ensure, or optimize for, smooth curvature between pre-trained and fine-tuned functions. We note that SAM-style algorithms could yield this assurance empirically with a large $\rho$ to always contain the trust region (i.e., Eq (2) > Eq (7) for all $\theta$ and $t$). However, we empirically observe that this is not the case — a larger setting of $\rho$ for SAM produces poorer results than TRAM. TRAM could be considered a subsolution of ASAM—our contribution highlights that sensitivity to the function space trust region produces better empirical outcomes with less catastrophic forgetting.
>
> [1] Better Fine-Tuning by Reducing Representational Collapse https://arxiv.org/abs/2008.03156
> [2] Natural Gradient Revisited https://openreview.net/forum?id=jbLdjjxPd-b2l
> [3] Trust Region Policy Optimization https://arxiv.org/abs/1502.05477
> [4] Sharpness-Aware Minimization for Efficiently Improving Generalization https://arxiv.org/pdf/2010.01412.pdf
> [5] Fisher SAM: Information Geometry and Sharpness Aware Minimisation https://proceedings.mlr.press/v162/kim22f/kim22f.pdf

---

> ### Author Response · Authors · 2023-11-21
> **Response Part 1B**
>
> **Q2: commentary on SWA**
> We thank the reviewer for highlighting this recent work. We did examine SWA in our early experiments but did not find the results to be very competitive with Trust region and SAM-style training in the cross-lingual transfer context. These results were omitted from the paper purely as we considered this additional discussion to be a distractor from the contrast between TRAM and the parent SAM and TR methods. Third party feedback has also requested this comparison and we will include this in future revisions.
>
> Here we show our results for SWA on XNLI compared to SAM/ASAM/FSAM and TRAM.
>
> | Algorithm               | {\sc ZS Avg} |
> |-------------------------|--------------|
> | Adam                    | 72.9         |
> | SAM                     | 73.7         |
> | ASAM                    | 74.0         |
> | FSAM                    | 73.8         |
> | **SWA**                     | 73.4         |
> | TRAM-$d_{x}$            | 75.2         |
> | TRAM-$d_{\theta_{t-1}}$ | 75.0         |
> | TRAM-$d_{\theta_{0}}$   | 74.9         |
> | TRAM-Fisher             | 74.4         |
>
>
> We identified several trends within our SWA results which warrant further study and we are actively exploring. The TL;DR of these findings is that weight averaging for optimal in-domain and out-of-domain performance is extremely variable across optimizers. We observe the same pattern as prior work in “turning on” SWA after some training, to average later checkpoints, works best for ID performance. However, we also identify that the best model for OOD uses only early checkpoints and training can stop after 30% of training. We also observe negligible correlation between performances for ID, OOD and where averaging starts/stops. We find SWA+SAM to perform below SAM alone but the inverse trend for Adam or SGD. We consider in-depth study of these phenomena as a different contribution which we plan to address in future work.
>
> **Q3: on the abstract claims**
>
> We thank the reviewer for their feedback on the abstract and claims. We will revise these for the scope of our claims to be clearer. We ask the reviewer if they can clarify where they consider more insight to be necessary.
>
> **Q4: Larger model scales**
>
> We agree with the reviewer that results on larger Llama scale models would be insightful. At present, we don’t have the resources to run full fine-tuning of larger models with TRAM.  In Appendix C.2, we highlight that training GPT2-XL (1.5B params) with TRAM yields a more generalizable model than other baseline algorithms including SAM, ASAM, FSAM and Trust Region methods. This is intended to provide some insight into if larger scales diminish the benefit of optimization contributions (where we find that TRAM is still beneficial at the 1.5B model scale). This echoes other recent work which finds that Sharpness-Aware optimization is still beneficial at many model scales [1]. We conjecture that TRAM would benefit a larger model as it has for 100M – 1.5B scales.
>
> [1] Sharpness-Aware Minimization Improves Language Model Generalization https://aclanthology.org/2022.acl-long.508/
>
> **Q5: Code release**
>
> We include an anonymised code repository for critical code elements of TRAM https://anonymous.4open.science/r/tram_optimizer-0448/tram.py.

---

> > ### Comment · Reviewer_2Ruf · 2023-11-22
> > **Thanks for the response.**
> >
> > I have read the response, and thanks for the authors' experiments. It is interesting to see the results regarding the SWA. And thanks for the interpretation, but it seems not entirely clear for me. I know it is hard to contribute on this point, but this warrants further research. I think the paper may contribute to SAM from some interesting persepctive. Although the rationality behind isn't completely articulated , I still consider the paper as a viable candidate for acceptance.

---

### Official Review · Reviewer_MZmv · 2023-11-01

**Soundness:** 3 good
**Presentation:** 3 good
**Contribution:** 3 good
**Rating:** 8
**Confidence:** 3

**Summary:**

This paper proposes a new optimization algorithm called TRAM that combines sharpness-aware minimization (SAM) with trust region regularization. SAM methods like ASAM optimize for low sharpness (flat minima) in parameter space. Trust region methods constrain optimization to a local neighborhood in representation space. TRAM unifies these approaches by bounding the SAM perturbation region using the trust region distance. This encourages flat minima while retaining representation smoothness.

**Strengths:**

1. The proposed method is intuitive and well-motivated. The combination of SAM and Trust region methods is reasonable and interesting.
2. Extensive experiments on multiple NLP tasks demonstrate the effectiveness of the proposed method.

**Weaknesses:**

1. Theoretical motivation for unifying SAM and Trust region methods is not provided.
2. Some results have high variance across runs. More runs may better characterize the performance.

**Questions:**

1. How well does TRAM transfer to other modalities like images?
2. There are several hyper-parameters of the proposed method. How to select them for out-of-distribution generalization?

---

> ### Author Response · Authors · 2023-11-22
> **Response Part 1**
>
> We thank the reviewer for their time and attention in reviewing our work. We now address each question in turn.
>
> **1. Theoretical Motivation**
>
> Please see our response to 2RUf Q1 for further insight on the notion of Trust and cqYa Q1 for discussion on the generalisation bound.
>
> **2. Run variance**
>
> We appreciate the concern for the run variation. At present, we do not have sufficient compute bandwidth for repeated runs of the LM objective but will endeavour for this in future revisions. All results for XNLI are averaged over 20 runs, which we consider sufficient for our claims and many more runs than is often claimed for experiments on cross-lingual transfer. The variance across different languages is more indicative of the task challenges than the run variance. We note that our vision experiments provided below do not report notably higher confidence intervals than other methods.
>
> **3. Vision modality**
>
> For comparison, we implement the same experiments as [1] for FisherSAM training ViT-base-16 [2] for CIFAR-100 [3], Flowers [4] and Cars [5]. We fine-tune ViT-base-16 for 200 epochs training with TRAM and report the Top-1 accuracy(+ 95% confidence interval) averaged over 5 runs  to compare directly to Table 3 in [1]. We match the hyperparameter setup of [1] — the base optimizer is SGD with an initial learning rate of 5e-4 and a cosine decay LR schedule. Due to computational constraints, we only have the bandwidth to report the two main variants of TRAM on these tasks during the rebuttal. We plan to update with the other variants of TRAM (TRAM-$d_{\theta_0}$ and TRAM-Fisher) in future revisions.
>
> |           | SGD        | SAM        | ASAM       | FSAM       | TRAM-$d_{\theta_{t-1}}$ | TRAM-$d_{x}$ |
> |-----------|------------|------------|------------|------------|-------------------------|--------------|
> | CIFAR-100 | 87.97±0.12 | 87.99±0.09 | 87.97±0.08 | 88.39±0.13 | 88.47±0.16              | **88.78**±0.01   |
> | Cars      | 92.85±0.31 | 93.29±0.01 | 93.28±0.02 | 93.42±0.01 | **93.49**±0.04              | 93.32±0.11   |
> | Flowers   | 94.53±0.20 | 95.05±0.06 | 95.08±0.10 | 95.26±0.03 | **97.07**±0.10              | 96.34±0.03   |
>
> We observe that TRAM performs competitively across all datasets, with one or both variants of TRAM performing above all other methods. The largest improvement for TRAM is Flowers where we perform +1.81% above FSAM. The smallest improvement is for Cars where we perform 0.07% above FSAM, but we not that the confidence intervals for these results do not overlap.
>
> [1] Fisher SAM: Information Geometry and Sharpness Aware Minimisation https://proceedings.mlr.press/v162/kim22f/kim22f.pdf
> [2] An Image is Worth 16x16 Words: Transformers for Image Recognition at Scale https://arxiv.org/abs/2010.11929
> [3] Learning Multiple Layers of Features from Tiny Images https://www.cs.toronto.edu/~kriz/learning-features-2009-TR.pdf
> [4] C3D Object Representations for Fine-Grained Categorization http://vision.stanford.edu/pdf/3drr13.pdf
> [5] Automated Flower Classification over a Large Number of Classes https://ieeexplore.ieee.org/document/4756141

---

> > ### Author Response · Authors · 2023-11-22
> > **Response Part 2**
> >
> > **4. Hyperparameter selection**
> >
> > All versions of TRAM include a small minimum Trust region distance, $\nu$. We include this to ensure that, in the event that the computed Trust region distance is 0, the ascent stage of the algorithm does not collapse to a redundant operation (i.e., a permutation of magnitude 0 is added to each parameter). We include this for assurance, but practically observe that $d> 0$ during training at all steps. As this is practically unused, we set this to a fixed arbitrarily small value ($\nu=10^{-5}$). We observe no penalty from including this term compared to training without $\nu$ where the Trust region distance could be zero.
> >
> > TRAM using the history of the previous step i.e., using distance $d_{\theta_{t-1}}$ introduces no additional hyperparameters beyond $\nu$. We considered a rescaling or normalization of $d_{\theta_{t-1}}$, e.g., with some additional scalar hyperparameter (e.g., $\alpha d_{\theta_{t-1}}$), but observed no empirical benefit to this with the inclusion of an additional term $\alpha$. The version of TRAM using sampled noise requires the hyperparameter $\sigma$, the variance of the zero-mean noise. We select this in a similar manner to https://openreview.net/pdf?id=OQ08SN70M1V using a linear search. This influences the magnitude of the additive noise to the inputs. The intuition (again borrowed from https://openreview.net/pdf?id=OQ08SN70M1V) is that additive noise simulates a permutation of the inputs which the model should be robust to. The trust region regularizer penalizes an arbitrary random small change in the function output space to encourage smaller update steps. As for TRAM using the history, we find that there is no benefit to any additional rescaling factors. In fact, we observe that TRAM is less sensitive to the setting of $\sigma$ than SAM or ASAM is to the similar setting of $\rho$.
> >
> > HPO specifically for OOD optimization is beyond our scope but an interesting future study. In general, we followed prior practice in cross-lingual transfer where we do not assume access to any other language/domain beyond the training set [1]. This reflects a real-world setting better where OOD or cross-lingual inference may happen after training. We have examined this to some degree and found that different hyperparameters influence the ID and OOD performance. We choose to present results which are specifically blind to tuning for OOD performance — the most extreme case to act as the OOD lower bound. Please see our response to 2Ruf on comparisons to SWA for more discussion in this area.
> >
> > [1] Zero-shot Cross-lingual Transfer is Under-specified Optimization https://aclanthology.org/2022.repl4nlp-1.25.pdf
> >
> > ---
> > We hope these experiments provide additional utility to the benefits of TRAM. We ask the reviewer to consider revising their score if these additional responses have sufficiently addressed your questions.

---

> > > ### Comment · Reviewer_MZmv · 2023-11-23
> > > **Thanks for the rebuttal**
> > >
> > > Thank the authors for addressing my concerns. The experiments on vision modality and the detailed explanation on hyper-parameter selection match my expectations. Therefore, I raise my score to 8.

---

### Official Review · Reviewer_AFpQ · 2023-11-09

**Soundness:** 3 good
**Presentation:** 3 good
**Contribution:** 3 good
**Rating:** 8
**Confidence:** 4

**Summary:**

This paper develops a new SAM-style optimizer for better representation learning (especially for language modeling).
The key idea is simple-to-state (although the proposed algorithms are a bit more complicated than the original SAM).
The authors attempt to combine the best of both worlds between SAM and Trust-region:
1. SAM encourages the solution to be "flat" in the parameter space (assuming that the flat minima is desired)
2. Trust-region methods encourages the representation to be "smooth" or to stay close to the good "pre-trained" model initialization.
In particular, with the Trust-region feature, the authors claim that the proposed method has the benefit of not forgetting task-agnostic representations from pre-trained model and also learning "smooth" representation which is good for the transferability of representations.

Also, building on more advanced SAM algorithms like ASAM and FSAM, the authors develop other variants of TRAM which are used for the experiments and show better performance than the previous approaches.

**Strengths:**

- The paper did a good job summarizing the existing approaches and how the proposed method builds on top of them.
- The experimental settings are detailed, and reasonable.
- Experiments seem quite comprehensive at least for the settings considered in this work.
- It's quite remarkable that the proposed methods achieve best performance across different fine-tuning tasks.

**Weaknesses:**

- See the question section below.

**Questions:**

- As far as I understood, the original SAM paper became popular because of its extensive experiments over standard benchmark datasets. In particular, I remember SAM achieving state-of-the art for various vision tasks (CIFAR, ImageNet etc...). Given that the effectiveness of SAM was first demonstrated on these benchmark tasks, **for future research I think it is required for follow-up works to sanity check the performance of the proposed methods on the same setting as the original SAM paper**. In particular, if the new methods end up giving a worse performance than SAM for the settings considered in the SAM paper, that would be an important information for practitioners.
(As I mentioned in the strength part, the authors' quite comprehensive experiments on the language modeling tasks of choice look great; however, since this work follows up on the original SAM paper, **some experiments that benchmark the new method against the original SAM on the task that SAM did great seems required**.)

- Also, it's great that the authors built their default algorithm on ASAM. But, given that SAM has been quite popular, as a reader, I'm quite curious how the SAM version of TRAM (instead of ASAM) performs. In particular, **is ASAM type of updates really necessary?**

- Given that the main motivation of this work is to develop an optimizer for learning good-representation, I think at least one experiment is needed for pre-training from scratch. In particular, do you think having a reasonable pre-trained model is necessary for TRAM to work well? As far as I know, it's still debated in the literature **whether SAM-type of updates are required in the beginning of the training or at the end of the training**. Some theoretical works have claimed that it's only effective at the end of the training (**as did in this work**), but empirical works also have claimed that it's required from the beginning of the training.

- In Figure 1, could you clarify what the negative and positive slopes are supposed to be interpreted as? Also I can't really understand how to interpret this plot.

- In Table 5, why didn't you present the statistics for the other two variants of TRAM?

I acknowledge the novelty of this work. However, given the extensive experiments in the previous works (e.g. original SAM work), in order to make a case about the effectiveness of the proposed methods, **I think some more "sanity-check" experiments are needed. Especially, because this work is empirical in nature.** I'm voting for weak accept at the moment, but I'll make the final decision based on how the authors address my questions.

---

> ### Author Response · Authors · 2023-11-21
> **Response Part 1**
>
> We thank the reviewer for their time and attention in considering TRAM. We provide the first part of our response to address your questions. We are still working on experiments for the vision domain (Q1) and will update our response when these are finalized.
>
> **Q2. "is ASAM type of updates really necessary?"**
>
> We thank the reviewer for correctly pointing out that TRAM can build on either ASAM or SAM. We pursue building on ASAM due to empirical benefits of ASAM vs SAM in our early experiments. We experimented with both techniques and found that just as ASAM generally outperforms SAM, TRAM building on ASAM outperforms TRAM built on SAM. We show XNLI accuracy for TRAM without and with adaptive scaling. This corresponds to TRAM using SAM or ASAM respectively. TRAM using SAM scaling still performs above most prior methods (SAM, ASAM, FSAM, TRPO, MESA) but below TRAM using ASAM.
>
> | Model                   | Without Adaptive Scaling (SAM) | With Adaptive Scaling (ASAM) |
> |-------------------------|--------------------------|-----------------------|
> | TRAM-$d_{x}$            | 74.7                     | 75.2                  |
> | TRAM-$d_{\theta_{t-1}}$ | 74.1                     | 75.0                  |
> | TRAM-$d_{\theta_{0}}$   | 74.1                     | 74.9                  |
>
>  **Q3: "whether SAM-type of updates are required in the beginning of the training or at the end of the training."**
>
> We thank the reviewer for highlighting the ongoing debate of when to apply SAM-style optimization.
> Given that “pretrain then fine-tune” is a standard paradigm in NLP, and more commonly in CV, we focus our efforts on applying TRAM here to learn a task-specific model. Our understanding of pre-training dynamics (in NLP) is that an algorithm requiring multiple forward and backward passes would be too expensive for large scale pre-training from scratch. Furthermore, the benefits of these steps may be weaker than simple additional pre-training on more data (i.e., the bitter lesson). We consider the value of TRAM greatest when adapting a task-agnostic model (or mostly task agnostic) to a task-specific model.  A principle of TRAM is to maintain generality from pre-training, therefore requiring some task-agnostic initial training to obtain a “trustable” original model.
>
> **Q4: Fig 1.**
>
> The slope of these curves represents the correlation between the training domain (ArXiv Math) and different groups of zero-shot evaluation domains. These curves represent the trend of domain correlation across all optimization algorithms (SAM-style, trust region and TRAM). For a positive correlation, a point above the curve indicates performing better than the trend. For a negative correlation, a point below the curve indicates a better performance than the trend. We do this to highlight that TRAM beats expectations of how negatively-correlated domain transfer impacts performance for hard OOD cases.
>
> **Q5: Table 5**
>
> Table 5 includes both “main” variants of TRAM which directly compute a TR distance. We omitted the other variants due to last minute space issues. We provide the numbers below and will include these updates in revisions. In general, TRAM-$d_{\theta_{0}}$ and TRAM-Fisher align with our trends comparing TRAM and other algorithms. All variants of TRAM have among the lowest sharpness and the highest CKA similarity across all variants.
>
> | XLM-R $\epsilon$-sharpness $\downarrow$ | EN   | ZS Avg.        |
> |-----------------------------------------|------|----------------|
> | Adam                                    | 2.16 | 1.98$\pm$~0.79 |
> | SAM                                     | 1.43 | 3.32$\pm$~0.96 |
> | ASAM                                    | 2.57 | 2.22$\pm$~0.79 |
> | FSAM                                    | 2.34 | 2.62$\pm$~0.29 |
> | TRPO                                    | 6.17 | 2.36$\pm$~1.02 |
> | R3F                                     | 6.22 | 2.56$\pm$~1.21 |
> | MESA                                    | 2.76 | 5.48$\pm$~0.75 |
> | TRAM-$d_{x}$                            | 0.61 | 1.49$\pm$~0.49 |
> | TRAM-$d_{\theta_{t-1}}$                 | 0.50 | 1.19$\pm$~0.38 |
> | TRAM-$d_{\theta_{0}}$                   | 0.75 | 1.92$\pm$~0.24 |
> | TRAM-Fisher                             | 1.67 | 2.02$\pm$~0.49 |
>
> | XLM-R CKA $\uparrow$    | EN   | ZS Avg.        |
> |-------------------------|------|----------------|
> | Adam                    | 0.69 | 0.44$\pm$~0.1  |
> | SAM                     | 0.69 | 0.42$\pm$~0.1  |
> | ASAM                    | 0.69 | 0.42$\pm$~0.1  |
> | FSAM                    | 0.73 | 0.48$\pm$~0.1  |
> | TRPO                    | 0.7  | 0.45$\pm$~0.1  |
> | R3F                     | 0.66 | 0.4$\pm$~0.1   |
> | MESA                    | 0.67 | 0.42$\pm$~0.1  |
> | TRAM-$d_{x}$            | 0.75 | 0.54$\pm$~0.11 |
> | TRAM-$d_{\theta_{t-1}}$ | 0.77 | 0.57$\pm$~0.1  |
> | TRAM-$d_{\theta_{0}}$   | 0.69 | 0.45$\pm$~0.1  |
> | TRAM-Fisher             | 0.72 | 0.49$\pm$~0.1  |

---

> ### Author Response · Authors · 2023-11-22
> **Response Part 2**
>
> **Q1. Sanity check experiments**
>
> For comparison, we implement the same experiments as [1] for FisherSAM training ViT-base-16 [2] for CIFAR-100 [3], Flowers [4] and Cars [5]. We fine-tune ViT-base-16 for 200 epochs training with TRAM and report the Top-1 accuracy(+ 95% confidence interval) averaged over 5 runs  to compare directly to Table 3 in [1]. We match the hyperparameter setup of [1] — the base optimizer is SGD with an initial learning rate of 5e-4 and a cosine decay LR schedule. Due to computational constraints, we only have the bandwidth to report the two main variants of TRAM on these tasks during the rebuttal. We plan to update with the other variants of TRAM (TRAM-$d_{\theta_0}$ and TRAM-Fisher) in future revisions.
>
> |           | SGD        | SAM        | ASAM       | FSAM       | TRAM-$d_{\theta_{t-1}}$ | TRAM-$d_{x}$ |
> |-----------|------------|------------|------------|------------|-------------------------|--------------|
> | CIFAR-100 | 87.97±0.12 | 87.99±0.09 | 87.97±0.08 | 88.39±0.13 | 88.47±0.16              | **88.78**±0.01   |
> | Cars      | 92.85±0.31 | 93.29±0.01 | 93.28±0.02 | 93.42±0.01 | **93.49**±0.04              | 93.32±0.11   |
> | Flowers   | 94.53±0.20 | 95.05±0.06 | 95.08±0.10 | 95.26±0.03 | **97.07**±0.10              | 96.34±0.03   |
>
>
> We observe that TRAM performs competitively across all datasets, with one or both variants of TRAM performing above all other methods. The largest improvement for TRAM is Flowers where we perform +1.81% above FSAM. The smallest improvement is for Cars where we perform 0.07% above FSAM, but we not that the confidence intervals for these results do not overlap.
>
>  - [1] Fisher SAM: Information Geometry and Sharpness Aware Minimisation https://proceedings.mlr.press/v162/kim22f/kim22f.pdf
>  - [2] An Image is Worth 16x16 Words: Transformers for Image Recognition at Scale https://arxiv.org/abs/2010.11929
>  - [3] Learning Multiple Layers of Features from Tiny Images https://www.cs.toronto.edu/~kriz/learning-features-2009-TR.pdf
>  - [4] C3D Object Representations for Fine-Grained Categorization http://vision.stanford.edu/pdf/3drr13.pdf
>  - [5] Automated Flower Classification over a Large Number of Classes https://ieeexplore.ieee.org/document/4756141
> ---
> We hope these experiments provide additional utility to the benefits of TRAM. We ask the reviewer to consider revising their score if these additional responses have sufficiently addressed your questions.

---

> > ### Comment · Reviewer_AFpQ · 2023-11-23
> > **Thank you for your response**
> >
> > I read through the responses and they sufficiently address my concerns.
> > Hence, I raise my score to 7 (it seems that there is no 7, so I marked my score to 8).
> > Please update the manuscript accordingly, I strongly believe that making those points will make the paper even stronger.
> > Thank you for taking time responding to my questions.

---

### Meta-Review · Area_Chair_SnBp · 2023-12-06

**Metareview:**

This paper proposes a SAM-type optimization that is tailored for fine-tuning tasks. Specifically, by imposing a trust region on top of seeking flat minima, the method enhances immunity against forgetting the original task and over fitting to the new task. The paper presents multiple variants of the proposed method and performs a reasonable empirical assessment on cross-domain language modeling and cross-lingual transfer, Per reviewers' request, the authors provided additional evaluation on vision tasks to ensure the gains are not limited to a specific modality. In addition, reviewers asked for some additional experiments around the difference between using SAM vs ASAM in the proposed method and comparison with other techniques believed to find flat minima such as SWA. Authors presented extra results about these in their response. As a result, two reviewers raised their score to 8 and the overall rating of the paper is clearly on the accept side. Reviewer cqYa who gave the lowest score (5) was seeking some clarity on how Trust Region method can improve transferability. Per this feedback, I suggest authors expand their discussion about the connection between transferibility and trust region, perhaps by including a few representative papers for further reading on this topic.

In concordance with majority of the reviews, I believe this submission has providing interesting insights at the intersection of transfer learning and SAM-type optimization, and I recommend accept. Please incorporate the (very helpful) information that you provided in your response, either into the main paper or an appendix.

**Justification For Why Not Higher Score:**

The proposed method adapts from established approaches.

**Justification For Why Not Lower Score:**

The empirical results are convincing and consider both language and vision modalities.

---

### Decision · Program_Chairs · 2024-01-16

Accept (spotlight)